# FAKE IT TILL MAKE IT: FEDERATED LEARNING WITH CONSENSUS-ORIENTED GENERATION

**Rui Ye[1,3], Yaxin Du[1,3], Zhenyang Ni[1,3], Yanfeng Wang[2,1], Siheng Chen[1,2,3]✉**
[1]Shanghai Jiao Tong University, [2]Shanghai AI Laboratory
[3]Multi-Agent Governance & Intelligence Crew (MAGIC)
{yr991129,dorothydu,0107nzy,wangyanfeng,sihengc}@sjtu.edu.cn

## ABSTRACT

In federated learning (FL), data heterogeneity is one key bottleneck that causes model divergence and limits performance. Addressing this, existing methods often regard data heterogeneity as an inherent property and propose to mitigate its adverse effects by correcting models. In this paper, we seek to break this inherent property by generating data to complement the original dataset to fundamentally mitigate heterogeneity level. As a novel attempt from the perspective of data, we propose federated learning with consensus-oriented generation (FedCOG). FedCOG consists of two key components at the client side: complementary data generation, which generates data extracted from the shared global model to complement the original dataset, and knowledge-distillation-based model training, which distills knowledge from global model to local model based on the generated data to mitigate over-fitting the original heterogeneous dataset. FedCOG has two critical advantages: 1) it can be a plug-and-play module to further improve the performance of most existing FL methods, and 2) it is naturally compatible with standard FL protocols such as Secure Aggregation since it makes no modification in communication process. Extensive experiments on classical and real-world FL datasets show that FedCOG consistently outperforms state-of-the-art methods. Code is available at https://github.com/rui-ye/FedCOG

## 1 INTRODUCTION

Federated learning (FL) is an emerging privacy-preserving training paradigm that enables multiple clients to train a global model collaboratively without directly accessing their raw data (McMahan et al., 2017). With the increasing privacy concerns and legislation, FL has attracted much attention from industry and academia (Kairouz et al., 2019; Yang et al., 2019). FL can be widely used in a diverse of real-world applications such as keyboard prediction (Hard et al., 2018; Yang et al., 2018).

As one of the most common and critical issues in FL, data heterogeneity fundamentally limits FL's practical performance (McMahan et al., 2017) as clients' datasets could be distinctly different from each other (Li et al., 2020a). To tackle this issue, enormous previous works focus on model-level correction at either the client or server side. At the client side, many methods propose to enhance the consistency among local models through regularization in model space (Li et al., 2020b) or feature space (Li et al., 2021a), or local gradient correction (Karimireddy et al., 2020). At the server side, some methods introduce update momentum (Hsu et al., 2019), apply knowledge distillation based on public dataset (Lin et al., 2020), or adjust aggregation manner (Wang et al., 2020b). However, these previous works regard the data heterogeneity as an *inherent property* and attempt to mitigate its negative effects via *model correction*, leaving the training adversely affected throughout the FL process as the inherent heterogeneity persists to cause client drift (Karimireddy et al., 2020).

In this paper, we seek an orthogonal approach to address the data heterogeneity issue: correcting the data itself to mitigate the heterogeneity level. To correct the data, our core idea is to generate data from the shared global model as consensus to complement the original data, which contributes to mitigate the effects of data heterogeneity by making all local datasets more homogeneous (e.g., local categorical distributions are more balanced).

Following this spirit of data correction, we propose a new FL algorithm, **Fed**erated Learning with **C**onsensus-**O**riented **G**eneration, denoted as **FedCOG**. FedCOG includes two novel components:

complementary data generation and knowledge-distillation-based model training. 1) During complementary dataset generation, each client generates data that is accurately predicted by global model but incorrectly predicted by local model. In this case, the generated data not only contains consensual knowledge in the global model but also serves as an informative dataset complement, mitigating the level of data heterogeneity. 2) During local model training, beside minimizing the conventional task-driven loss on the original dataset, each client distills the knowledge from global model to current local model based on the generated dataset. Such knowledge distillation contributes to enhance the consensus among local models, which further alleviates the impact of data heterogeneity. Overall, FedCOG improves the performance by mitigating both data heterogeneity level and its effects.

FedCOG has two critical properties: 1) plug-and-play property and 2) compatibility with conventional real-world FL protocol. First, FedCOG focuses on data correction, which is orthogonal to most existing FL methods (e.g., those that focus on model correction (Li et al., 2020b; Karimireddy et al., 2020)) and thus can be easily combined with them to further enhance the performance. Second, FedCOG is compatible with practical standard FL protocols such as Secure Aggregation (Bonawitz et al., 2017) since it does not modify the communication process on standard FL, making it convenient to deploy in real-world application.

With extensive experiments across different representative datasets including real-world multi-label FL dataset FLAIR (Song et al., 2022), heterogeneity types and heterogeneity levels, we show that our proposed FedCOG consistently outperforms 11 representative baselines and is of plug-and-play property. Besides, we conduct further empirical analysis to provide more insights in FedCOG from the perspective of alleviated model dissimilarity and enhanced local model performance.

The main contribution is as follows:

- We propose a novel attempt to tackle data heterogeneity in FL from the perspective of data correction, which seek to mitigate its adverse effects from the dataset itself.
- We propose a novel algorithm, FedCOG, which mitigates the issue of data heterogeneity by generating data to complement original data and distilling knowledge from global model to local model through the generated data.
- With extensive experiments, we show FedCOG consistently achieves the best and helps relieve model dissimilarity and forgetting.

## 2 RELATED WORK

**Data Heterogeneity in Federated Learning.** Data heterogeneity is one of the most fundamental and essential challenges in FL (McMahan et al., 2017; Kairouz et al., 2019), where data distributions of clients could distinctly vary from each other, and is shown to affect the performance of FL empirically (Zhao et al., 2018; Li et al., 2020b) and theoretically (Li et al., 2019; Yu et al., 2019). Addressing this, many methods are proposed from the aspects of local model correction and global model adjustment. **1) Local model correction** aims for more similar local models at the client side. FedProx (Li et al., 2020b) and FedDyn (Acar et al., 2020) propose regularizing the distance between local and global model. SCAFFOLD (Karimireddy et al., 2020) introduces control variate to correct local gradients, MOON (Li et al., 2021a), FedFM (Ye et al., 2023b) and FedDecorr (Shi et al., 2022) regularize the local models from the feature space via feature alignment or feature correlation regularization. **2) Global model adjustment** aims for better-performed global model at the server side. FedAvgM (Hsu et al., 2019) and FedOPT (Reddi et al., 2021) introduce momentum to stabilize global model updating. FedNova (Wang et al., 2020b) and FedDisco (Ye et al., 2023c) modify the aggregation rules by adjusting aggregation weights. Others may apply knowledge distillation based on a public dataset (Lin et al., 2020) or design client sampling strategies (Cho et al., 2020).

Different from these two aspects, our method addresses this issue in a novel way by directly reducing heterogeneity level from the aspect of data. We propose to generate additional data samples to complement the original heterogeneous dataset for each client, which is achieved by inversely optimizing inputs given the global and local model.

**Data Generation and Augmentation in FL.** (1) Recently, some methods (Zhang et al., 2022a;b; Zhu et al., 2021; Hao et al., 2021) apply data generation at the server side to tune the global model for better performance, which generate data based on all local models. However, this could be un-

acceptable in practice since Secure Aggregation technique (Bonawitz et al., 2017; So et al., 2022) is often required so that the server can only receive the aggregated global model, making these methods less practical and privacy-preserving. In contrast, our proposed FedCOG is compatible with Secure Aggregation and fundamentally mitigates the data heterogeneity by generating complementary data. (2) Some methods (Tang et al., 2022; Chen et al., 2022; Yuan et al., 2022) apply data augmentation, though, they all introduce additional communication objects other than model parameters (McMahan et al., 2017), which increases communication cost and/or risks of privacy leakage. Others (Xu et al., 2022; Yoon et al., 2021) may generate data to address the problem of forgetting. Unlike these, FedCOG makes no compromise on privacy and communication cost, and is a novel method that adopts task-specific data generation to tackle data heterogeneity.

**Data Generation from Model.** There are a line of works that focus on data generation from a single model (Fredrikson et al., 2015; Mahendran & Vedaldi, 2015; 2016) to understand the representations of neural networks. Some works extend data generation to achieve data-free knowledge distillation (Fang et al., 2019; Nayak et al., 2019; Mordvintsev et al., 2015; Yin et al., 2020). For example, DeepDream (Mordvintsev et al., 2015) generates data by penalizing classification loss, total variance and $\ell_2$ norm while DeepInversion (Yin et al., 2020) additionally adds a feature regularization and divergence term. These works are orthogonal to ours as they do not consider FL scenarios and our method can be further enhanced by combining with more advanced generation techniques.

## 3 PROBLEM FORMULATION

The objective of federated learning (FL) is to enable multiple clients collaboratively train a global model under the coordination of a server without directly accessing raw data (Wang et al., 2021). Specifically, suppose that there are $K$ clients, where each client $k$ holds a private local original dataset $\mathcal{D}_k$, the global objective of FL is: $\min_{\boldsymbol{\theta}} \sum_{k=1}^{K} p_k \mathcal{L}_c(\boldsymbol{\theta}; \mathcal{D}_k)$, where $p_k = \frac{|\mathcal{D}_k|}{\sum_i |\mathcal{D}_i|}$ is the relative dataset size, $\mathcal{L}_c(\cdot)$ is the task-driven loss function. Conventionally, this optimization problem is solved by the two iterative steps, including 1) client step: client downloads global model $\boldsymbol{\theta}$ from the server and conducts training on local dataset $\mathcal{D}_k$ to obtain local model $\boldsymbol{\theta}_k$, 2) server step: server receives local models $\{\boldsymbol{\theta}_k\}$ from clients and aggregates them to obtain global model $\boldsymbol{\theta} := \sum_k p_k \boldsymbol{\theta}_k$.

Since clients' datasets $\{\mathcal{D}_k\}$ can be heterogeneous, the trained local models at client step can be significantly different and eventually results in slow convergence and limited performance of global model. Some methods are proposed to mitigate the difference between local models $\{\boldsymbol{\theta}_k\}$ at client step (Li et al., 2020b; Karimireddy et al., 2020) while some to improve the global model $\boldsymbol{\theta}$ at the server step (Reddi et al., 2021; Jhunjhunwala et al., 2023).

However, they do not consider modification on the original datasets $\{\mathcal{D}_k\}$, which is the root cause of heterogeneity issue. In this paper, we propose a novel attempt for mitigating the effects of data heterogeneity by generating data to complement the original dataset for each client.

## 4 METHODOLOGY

We propose federated learning with consensus-oriented generation (FedCOG), which mitigates data heterogeneity by generating complementary data and refining training at the client side. FedCOG follows the standard FL protocol that consists of client and server steps (McMahan et al., 2017). The novel designs of FedCOG lie in the **client step**, which is composed of two key modules: generating data to complement the original dataset and conducting knowledge distillation on the local model, which are elaborated in the following; also see details in Algorithm 1 and illustration in Figure 1.

### 4.1 COMPLEMENTARY DATASET GENERATION

As a novel attempt to mitigate the adverse effects of data heterogeneity from the perspective of data, the key step in FedCOG is generating data to complement the original dataset of each client, reducing the heterogeneity levels. We propose to generate task-specific and client-specific data by learning to produce samples that can be accurately predicted by the current global model and falsely predicted by the previous local model. Such generated data can contain consensual knowledge for the targeted task and complement the client dataset to increase data diversity.

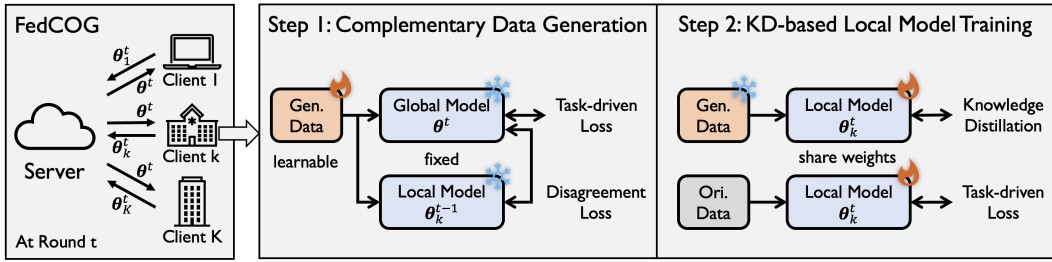

Figure 1: Overview of FedCOG, which consists of two steps at the client side. 1) Complementary data generation by optimizing the inputs, supervised by task-driven and disagreement loss. 2) Knowledge-distillation-based local model training by distilling knowledge from global model to local model through generated data and minimizing task-driven loss through original data.

To obtain the desired samples, we set the inputs of the model as learnable parameters and fix the model parameters of global model and local model, where the learnable inputs are optimized by the guidance of task-driven loss and disagreement loss. Specifically, let $\boldsymbol{\theta}^t$ be the global model at round $t$ and $\boldsymbol{\theta}_k^{t-1,r}$ be client $k$'s local model at previous round $t-1$ after $r$ SGD iterations of local model training. We use $\boldsymbol{\theta}^{*|t}$ to represent that the model parameters are fixed and $\boldsymbol{\theta}^t$ to represent that the model parameters are learnable. Denote $\mathcal{D}_k = \{(\hat{\boldsymbol{x}}_i, y_i)|i \in N\}$ as the generated dataset of client $k$ where $N$ is the number of samples to generate, $\hat{\boldsymbol{x}}_i \in \mathbb{R}^{H \times W \times C}$ ($H, W, C$ denotes height, width and number of channels) is a learnable tensor and $y_i$ is an arbitrary pre-defined target label. Each client $k$ generates data by solving the following optimization problem:

$$\widehat{\mathcal{D}}_k^t := \underset{\mathcal{D}_k}{\arg\min}\, \mathcal{L}_{gen}(\boldsymbol{\theta}^{*|t}, \boldsymbol{\theta}_k^{*|t-1,\tau}; \mathcal{D}_k) = \frac{1}{|\mathcal{D}_k|} \sum_{(\hat{\boldsymbol{x}},y) \in \mathcal{D}_k} \mathcal{L}_c(\boldsymbol{\theta}^{*|t}; \hat{\boldsymbol{x}}, y) + \lambda_{dis}\mathcal{L}_{dis}(\boldsymbol{\theta}^{*|t}, \boldsymbol{\theta}_k^{*|t-1,\tau}; \hat{\boldsymbol{x}}),$$
(1)

where $\mathcal{L}_{gen}(\cdot)$ is the overall generating objective. $\mathcal{L}_c(\cdot)$ is the task-driven loss, $\lambda_{dis}$ is a hyper-parameter, and $\mathcal{L}_{dis}(\cdot)$ is the defined disagreement loss. Taking single-label classification task as a concrete example, we define the task-driven loss as a cross-entropy loss and the disagreement loss based on Jensen-Shannon divergence (Lin, 1991; Yin et al., 2020), which is formulated as:

$$\mathcal{L}_{dis}(\boldsymbol{\theta}^{*|t}, \boldsymbol{\theta}_k^{*|t-1,\tau}; \hat{\boldsymbol{x}}) := 1 - \frac{1}{2}\Big(KL\big(\boldsymbol{p}(\boldsymbol{\theta}^{*|t}; \hat{\boldsymbol{x}}), \overline{\boldsymbol{p}}\big) + KL\big(\boldsymbol{p}(\boldsymbol{\theta}_k^{*|t-1,\tau}; \hat{\boldsymbol{x}}), \overline{\boldsymbol{p}}\big)\Big),$$
(2)

where $\boldsymbol{p}(\boldsymbol{\theta}; \hat{\boldsymbol{x}})$ denotes the output of model $\boldsymbol{\theta}$, $\overline{\boldsymbol{p}} = \frac{1}{2}\big(\boldsymbol{p}(\boldsymbol{\theta}^{*|t}; \hat{\boldsymbol{x}}) + \boldsymbol{p}(\boldsymbol{\theta}_k^{*|t-1,\tau}; \hat{\boldsymbol{x}})\big)$ is the averaged output, and $KL(\cdot)$ is the Kullback–Leibler divergence (Kullback & Leibler, 1951).

The proposed two terms contribute to generating consensual and reasonable data to diversify and complement the local client dataset. 1) The first task-driven loss encourages generating data that is accurately recognized by the global model. Since the global model is the knowledge aggregation of multiple local models, such generated data is task-specific and can be seen as consensual knowledge of all clients. 2) The second disagreement term encourages the generated data to cause global-local model disagreement, serving as a better complement for the original client dataset as the generated data is generally 'unfamiliar' for the client's local model. The generated dataset $\widehat{\mathcal{D}}_k$ is then utilized in local model training. See discussions on the applicability to other modalities in Appendix A.1.2.

**Details of designing target labels for generated data.** One basic generating strategy is uniformly generating data for all possible target labels. Taking CIFAR-10 as a concrete task example, the target label list can be $[\underbrace{0, ..., 0}_{n}, \underbrace{1, ..., 1}_{n}, ..., \underbrace{9, ..., 9}_{n}]$, resulting in $10 \times n$ samples to generate. Though, there can also be more specific designs for particular tasks; see Appendix A.1.1.

## 4.2 LOCAL MODEL TRAINING WITH KNOWLEDGE DISTILLATION

After data generation, the pivotal next step involves incorporating the generated data into local model training. A simple and straightforward solution is to merge the generated dataset with the original dataset, creating a new dataset for local model training. However, since there are inherent discrepancies between generated and real-world data, direct training on the generated data in a hard-labeled

---

**Algorithm 1** FedCOG: federated learning with consensus-oriented generation.
1: **Initialization:** number of clients $K$, number of rounds $T$, number of iterations $\tau$.
2: **for** $t = 0, 1, \ldots, T - 1$ **do**             ▷ FL rounds in sequence
3:      Send global model $\boldsymbol{\theta}^t$ to each client           ▷ Model communication
4:      **for** $k = 0, 1, \ldots, K - 1$ in parallel **do**          ▷ Clients in parallel
5:          $\widehat{\mathcal{D}}_k \leftarrow \arg\min_{\mathcal{D}} \mathcal{L}_{gen}(\boldsymbol{\theta}^{*|t}, \boldsymbol{\theta}_k^{*|t-1,\tau}; \mathcal{D})$     ▷ Complementary data generation
6:          $\boldsymbol{\theta}_k^{t,0} := \boldsymbol{\theta}^t$             ▷ Model synchronization
7:          **for** $r = 0, 1, \ldots, \tau - 1$ **do**           ▷ Iterations in sequence
8:             $\boldsymbol{\theta}_k^{t,r+1} := \boldsymbol{\theta}_k^{t,r} - \eta\nabla\big(\mathcal{L}_c(\boldsymbol{\theta}_k^{t,r}) + \lambda_{kd}\mathcal{L}_{kd}(\boldsymbol{\theta}^{*|t}, \boldsymbol{\theta}_k^{t,r})\big)$    ▷ KD-based model training
9:      Send local model $\boldsymbol{\theta}_k^{t,\tau}$ to server          ▷ Model communication
10:     $\boldsymbol{\theta}^{t+1} := \sum_{k=1}^{K} p_k \boldsymbol{\theta}_k^{t,\tau}$            ▷ Model aggregation
11: **Return:** $\boldsymbol{\theta}^T$

---

format may result in overly stringent supervision. Thus, we advocate using the global model's soft labels for guidance, striking a balance between learning from valuable insights from the generated data while circumventing the potential biases from the generation process.

Specifically, client $k$ initially employs the global model to reinitialize the local model: $\boldsymbol{\theta}_k^{t,0} := \boldsymbol{\theta}^t$, and subsequently launches local model training on both of the original dataset $\mathcal{D}_k$ and the generated dataset $\widehat{\mathcal{D}}_k$, with the optimization objective defined as:

$$\min_{\boldsymbol{\theta}} \frac{1}{|\mathcal{D}_k|} \sum_{(\boldsymbol{x}_i, y_i) \in \mathcal{D}_k} \mathcal{L}_c(\boldsymbol{\theta}; \boldsymbol{x}_i, y_i) + \frac{\lambda_{kd}}{|\widehat{\mathcal{D}}_k|} \sum_{(\hat{\boldsymbol{x}}_i, y_i) \in \widehat{\mathcal{D}}_k} KL\big(\boldsymbol{p}(\boldsymbol{\theta}^{*|t}; \hat{\boldsymbol{x}}), \boldsymbol{p}(\boldsymbol{\theta}; \hat{\boldsymbol{x}})\big), \quad (3)$$

where $\boldsymbol{p}(\boldsymbol{\theta}^{*|t}; \hat{\boldsymbol{x}})$ is the output of global model, $\boldsymbol{p}(\boldsymbol{\theta}; \hat{\boldsymbol{x}})$ is the output of currently optimizing local model, and $\lambda_{kd}$ is a hyper-parameter. Note that the outputs of the global model given generated data can be extracted from the previous generation step without the need for recomputation. Solving the optimization problem for $\tau$ steps of standard SGD in total, each client obtains the trained local model $\boldsymbol{\theta}_k^{t,\tau}$, which is then uploaded to the server for further model aggregation.

In this objective function, the optimization of the local model is balanced between task-driven loss (optimizing on the original real-world dataset) and knowledge distillation loss (optimizing on the generated dataset). This approach serves two purposes: first, to prevent over-fitting on the locally heterogeneous original dataset; and second, to preserve the knowledge of the global model and prevent excessive loss of consensus knowledge during local training.

## 4.3 DISCUSSIONS

**Convergence analysis.** We provide a theoretical convergence analysis in Appendix A.5.

**Compatibility.** FedCOG is compatible with standard FL communication protocols including Secure Aggregation (SA) (Bonawitz et al., 2017) since it does not modify the communication process compared with standard FL (McMahan et al., 2017). However, many previous works are not compatible with SA since they assume that the server can directly access individual local models (Lin et al., 2020; Zhang et al., 2022a;b); while SA requires that the server can only obtain the summation of local models to enhance security and privacy (Zhu et al., 2019). Besides, FedCOG has the plug-and-play property that can be combined with many existing FL works (Hsu et al., 2019) and can be potentially applied in future works performance improvement; see empirical evidence in Table 3.

**Communication, privacy, and computation.** As pointed in (Kairouz et al., 2019), privacy and communication efficiency are two first-order concerns in FL, our proposed FedCOG does not compromise on either of these two aspects since it does not introduce any additional communication objects other than model parameters as FedAvg (McMahan et al., 2017). As there is no free lunch for privacy, utility, and efficiency in FL (Zhang et al., 2022c), we search to enhance utility by slightly sacrificing computation efficiency, which is a relatively minor concern in FL. Though, the introduced computation overhead is acceptable as 1) the introduced generator is lightweight; see the comparison of training cost in Table 4; and 2) generating data for only a few rounds is adequate to bring evident performance improvement; see experiments in Table 3. We provide further discussions in A.5.2.

Table 1: Accuracy (%) comparisons across three datasets and two heterogeneous scenarios. The last column records the averaged accuracy across settings and FedCOG achieves the highest accuracy.

| Dataset | Fashion-MNIST | | CIFAR-10 | | CIFAR-100 | | Avg. |
|---|---|---|---|---|---|---|---|
| Heterogeneity | NIID-1 | NIID-2 | NIID-1 | NIID-2 | NIID-1 | NIID-2 | |
| FedAvg | $73.07_{\pm0.08}$ | $64.11_{\pm0.78}$ | $64.36_{\pm0.11}$ | $50.55_{\pm0.45}$ | $39.07_{\pm0.36}$ | $30.93_{\pm0.23}$ | 53.68 |
| FedAvgM | $76.81_{\pm0.48}$ | $67.73_{\pm0.65}$ | $64.36_{\pm0.33}$ | $51.32_{\pm0.74}$ | $39.21_{\pm0.19}$ | $30.97_{\pm0.11}$ | 55.07 |
| FedProx | $73.35_{\pm0.07}$ | $64.44_{\pm1.36}$ | $63.91_{\pm0.36}$ | $53.51_{\pm0.32}$ | $38.81_{\pm0.03}$ | $30.76_{\pm0.21}$ | 54.13 |
| SCAFFOLD | $77.02_{\pm0.63}$ | $58.47_{\pm3.05}$ | $64.36_{\pm0.43}$ | $48.35_{\pm1.86}$ | $42.25_{\pm0.40}$ | $34.27_{\pm0.22}$ | 54.12 |
| MOON | $73.21_{\pm0.28}$ | $63.11_{\pm1.08}$ | $63.65_{\pm0.04}$ | $51.01_{\pm1.54}$ | $39.34_{\pm0.18}$ | $31.17_{\pm0.13}$ | 53.58 |
| FedSAM | $76.44_{\pm0.06}$ | $68.14_{\pm0.60}$ | $61.38_{\pm0.23}$ | $50.76_{\pm0.36}$ | $37.97_{\pm0.26}$ | $30.75_{\pm0.11}$ | 54.24 |
| VHL | $73.08_{\pm0.89}$ | $70.26_{\pm1.79}$ | $60.81_{\pm1.33}$ | $51.60_{\pm0.98}$ | $37.55_{\pm1.06}$ | $28.41_{\pm0.48}$ | 53.62 |
| FedExP | $65.03_{\pm1.14}$ | $59.49_{\pm1.69}$ | $48.96_{\pm2.11}$ | $37.97_{\pm1.75}$ | $35.14_{\pm1.25}$ | $22.47_{\pm0.29}$ | 44.84 |
| FedDecorr | $72.77_{\pm0.70}$ | $60.70_{\pm4.59}$ | $63.40_{\pm0.29}$ | $47.53_{\pm1.99}$ | $37.68_{\pm0.40}$ | $27.08_{\pm0.29}$ | 51.53 |
| FedReg | $70.75_{\pm0.23}$ | $69.89_{\pm0.74}$ | $55.40_{\pm0.33}$ | $46.64_{\pm0.42}$ | $30.95_{\pm0.39}$ | $23.70_{\pm0.31}$ | 49.52 |
| FedGen | $72.64_{\pm1.27}$ | $61.34_{\pm2.44}$ | $62.45_{\pm0.40}$ | $49.12_{\pm0.51}$ | $37.99_{\pm0.23}$ | $26.93_{\pm0.15}$ | 51.75 |
| **FedCOG (Ours)** | $\mathbf{77.34}_{\pm0.07}$ | $\mathbf{73.68}_{\pm0.38}$ | $\mathbf{64.83}_{\pm0.12}$ | $\mathbf{54.00}_{\pm0.84}$ | $\mathbf{42.88}_{\pm0.02}$ | $\mathbf{34.80}_{\pm0.42}$ | **57.92** |

## 5 EXPERIMENTS

We conduct comprehensive comparisons with many FL baselines on multiple datasets, including real-world FL dataset FLAIR (Song et al., 2022). See more details and results in Section A.

### 5.1 IMPLEMENTATION DETAILS

**Heterogeneous datasets.** Overall, we consider three classical datasets and one real-world FL dataset. 1) For the three classical datasets, including Fashion-MNIST (Xiao et al., 2017), CIFAR-10/100 (Krizhevsky et al., 2009), we consider two types of heterogeneity, namely NIID-1 and NIID-2. NIID-1 follows Dirichlet distribution $Dir_\beta$ (default $\beta = 0.1$), which is a common setting in FL (Yurochkin et al., 2019; Wang et al., 2020a). NIID-2 makes each client hold data of only partial labels (McMahan et al., 2017; Li et al., 2020b) (2 for Fashion-MNIST and CIFAR-10, 10 for CIFAR-100). 2) FLAIR (Song et al., 2022) is a recently released real-world FL multi-label dataset, where each client is a user from Flickr and we adopt the task with 17 labels. It is a challenging task since each user has different data distribution and few data samples. See more in Section A.3.

**Training setups.** For the three classical datasets, we consider 70 communication rounds, $K = 10$ clients, $\tau = 400$ iterations of model training, and apply a simple CNN (Li et al., 2021a). For the more challenging FLAIR (Song et al., 2022) dataset, we consider 400 communication rounds and apply ResNet18 (He et al., 2016) model. Besides, we sample $K = 200$ clients in total and 10 clients in each FL round, and set $\tau = 10$ because each client holds relatively few data samples. For all datasets, we use SGD as the optimizer with learning rate 0.01. For FedCOG, we apply the most computation-efficient way by setting the inputs as learnable, supervised by 256 pre-defined target labels. The default settings of $\lambda_{dis}$ and $\lambda_{kd}$ are 0.1 and 0.01, respectively.

**Baselines.** We compare with 11 baselines. FedAvg (McMahan et al., 2017) is the basical baseline. FedProx (Li et al., 2020b), SCAFFOLD (Karimireddy et al., 2020), MOON (Li et al., 2021a), Fed-SAM (Qu et al., 2022), FedDecorr (Shi et al., 2022) focus on local model correction, VHL (Tang et al., 2022), FedReg (Xu et al., 2022) and FedGen (Zhu et al., 2021) focus on data augmentation, FedAvgM (Hsu et al., 2019) and FedExP (Jhunjhunwala et al., 2023) focus on global model adjustment. Among these, SCAFFOLD requires double communication cost.

### 5.2 COMPARISONS WITH STATE-OF-THE-ART METHODS

**Applicability on standard datasets.** Here, we conduct experiments under two types of heterogeneous scenarios on three standard datasets used in FL literature (Li et al., 2020b; 2021a; Shi et al., 2022; Jhunjhunwala et al., 2023), including Fashion-MNIST (Xiao et al., 2017), CIFAR-10/100 (Krizhevsky et al., 2009). We run 50 rounds of FedAvg before running 20 rounds of Fed-COG for higher computation efficiency. We report accuracy comparisons in Table 1. From the table, we see that 1) FedCOG consistently performs the best across different datasets and heterogeneous scenarios, indicating the effectiveness of tackling data heterogeneity from the perspective of data. 2) For the relatively more challenging scenario, NIID-2, some methods (e.g., SCAFFOLD) even

Table 3: Efficient plug-and-play property. FedCOG consistently brings performance improvement across different scenarios, with only 1 round for CIFAR-10 and 3 rounds for FLAIR.

| Method | FedAvg | | FedAvgM | | FedProx | | SCAFFOLD | | MOON | |
| + FedCOG? | × | ✓ | × | ✓ | × | ✓ | × | ✓ | × | ✓ |
|---|---|---|---|---|---|---|---|---|---|---|
| CIFAR-10-1 | 63.34 | **64.47** | 61.71 | **64.63** | 64.15 | **64.87** | 63.67 | **65.36** | 62.56 | **63.24** |
| CIFAR-10-2 | 49.30 | **49.50** | 44.73 | **45.14** | 50.35 | **50.89** | 46.47 | **51.26** | 49.64 | **50.72** |
| FLAIR O-F1 | 47.91 | **51.35** | 48.77 | **51.60** | 45.71 | **54.57** | 42.50 | **45.22** | 44.99 | **52.78** |
| FLAIR C-F1 | 20.63 | **20.89** | 20.11 | **21.98** | 18.66 | **19.08** | 16.74 | **18.96** | 18.83 | **19.92** |

perform worse than FedAvg; while in contrast, our proposed FedCOG brings significantly larger performance gain. This indicates that complementary data generation is more effective in cases where local dataset misses data from several categories (i.e., more heterogeneous).

**Applicability on the real-world FL multi-label dataset, FLAIR** (Song et al., 2022). We run additional 5 rounds of FedCOG based on the global model trained by 400 rounds of FedAvg, for the sake of computation efficiency. We compared with other baselines with 405 rounds for fair comparison by evaluating per-class (denoted by C-) precision, recall, F1 score, and per-sample (denoted by O-) F1 score in Table 2. From the table, we see that 1) FedCOG consistently achieves the best performance under different evaluation metrics, indicating the effectiveness of data generation in Fed-COG for multi-label task. 2) While some methods perform worse than the vanilla FL method FedAvg or even inapplicable to this task (e.g., VHL (Tang et al., 2022) and FedReg (Xu et al., 2022)), our proposed FedCOG even brings larger performance improvement for such real-world multi-label task.

Table 2: Experiments on real-world FL multi-label dataset, FLAIR. C- and O- denote averaged metric across 17 labels and samples, respectively. P: precision, R: recall.

| Method | C-P | C-R | C-F1 | O-F1 |
|---|---|---|---|---|
| FedAvg | 30.76 | 14.23 | 18.30 | 44.53 |
| FedAvgM | 29.76 | 16.76 | 20.44 | 48.43 |
| FedProx | 32.27 | 14.95 | 18.66 | 45.71 |
| SCAFFOLD | 31.10 | 12.37 | 16.74 | 42.50 |
| MOON | 31.41 | 14.71 | 18.83 | 44.99 |
| FedSAM | 32.33 | 14.39 | 17.94 | 46.18 |
| VHL | N/A | N/A | N/A | N/A |
| FedDecorr | 33.44 | 13.36 | 17.51 | 41.89 |
| FedExP | 29.14 | 15.70 | 19.64 | 46.57 |
| FedReg | N/A | N/A | N/A | N/A |
| **FedCOG** | **37.87** | **18.78** | **20.89** | **51.35** |

Specifically, it outperforms FedAvg by 14% and 15% relatively with regard to C-F1 and O-F1.

## 5.3 EFFICIENT PLUG-AND-PLAY PROPERTY

One valuable advantage of FedCOG is its plug-and-play nature since FedCOG focuses on data-level modification, which is orthogonal to most existing methods. In this section, we select five representative baseline methods, including FedAvg (McMahan et al., 2017), FedAvgM (Hsu et al., 2019), FedProx (Li et al., 2020b), SCAFFOLD (Karimireddy et al., 2020) and MOON (Li et al., 2021a), to investigate the compatibility of FedCOG with existing methods.

Here, we first conduct 50 rounds of training using the baseline method X. Subsequently, on one hand, we continue training method X for 1 round to obtain Model A, and on the other hand, we conduct 1 round of training of method X+FedCOG to obtain Model B. Finally, we compare the performances of Model A and B in Table 3. From the table, we see that 1) FedCOG consistently brings performance gain when combined with these baseline methods, indicating its plug-and-play property. Specifically, it can bring 19% O-F1 improvement to FedProx (Li et al., 2020b) on FLAIR dataset. 2) there is already evident performance improvement through only 1 round of FedCOG, indicating that FedCOG can take effects in a computation-efficient way.

## 5.4 EMPIRICAL ANALYSIS OF FEDCOG

We dive into the training process by evaluating model difference, generalization, and personalization of local models to provide more deeper insights. Experiments are conducted on a common setting (Shi et al., 2022; Jhunjhunwala et al., 2023), NIID-1 on CIFAR-10. We start from the same global model initialization (trained by 50 rounds of FedAvg (McMahan et al., 2017)), then launches 1 round of FedAvg, FedProx (Li et al., 2020b), and FedCOG, respectively. See more in Section A.5.

**Model difference of local models.** Here, we measure the $\ell_2$ difference between each local model and the aggregated global model, which is a reasonable indicator of the effects of data heterogene-

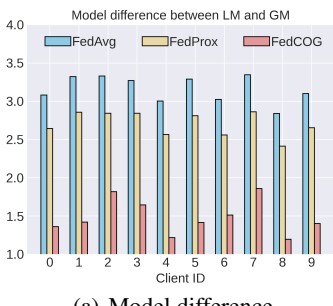 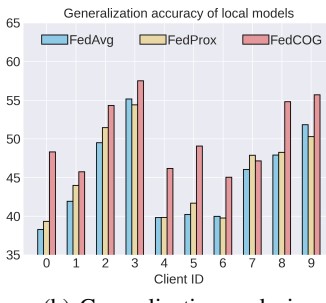 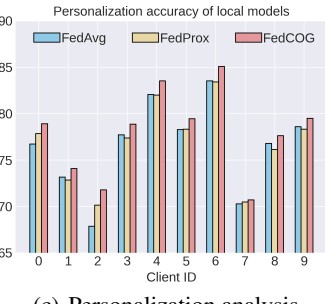

| (a) Model difference | (b) Generalization analysis | (c) Personalization analysis |
|---|---|---|

Figure 2: Empirical analysis of FedAvg, FedProx, and FedCOG. a) FedCOG has the lowest model difference between local model (LM) and global model (GM). b) FedCOG consistently and significantly achieves the best preservation of the generalization ability of local models. c) Interestingly, FedCOG also consistently achieves best personalization performance.

ity (Li et al., 2020b). From Figure 2(a), we see that FedCOG achieves the lowest model difference between local and global model, indicating that FedCOG can significantly reduce the effects of data heterogeneity on the consistency of local models.

**Generalization ability of local models.** Here, we evaluate each local model on the uniform test dataset (used in Table 1) to examine the preservation of general knowledge during local training. From Figure 2(b), we see that FedCOG consistently and significantly achieves the highest accuracy of local models, indicating that FedCOG preserves the generalization ability best since it generates data from global model to complement the original dataset. Additionally, the achieved global model accuracies of FedAvg, FedProx and FedCOG are 63.34, 64.57 and 65.59, respectively, indicating that FedCOG achieves best generalization performance of both local models and global model.

**Personalization ability of local models.** Here we evaluate each local model on the personalized test dataset, which follows the data distribution of its training dataset. Interestingly, from Figure 2(c), we find that FedCOG also consistently achieves the highest personalization performance, indicating that FedCOG can also enhances the fitting ability on local dataset. Such a performance gain of FedCOG could be attributed to two aspects: 1) From a global view, FedCOG leverages data generation and knowledge distillation to achieve a well-performing aggregated global model. This global model is then used as the initialization for local model training, contributing to improved performance. 2) From a local perspective, the generated data serves two purposes: it enhances model learning by providing additional knowledge, and it mitigates the risk of over-fitting. See the further explanations and comparisons of FedCOG against several representative personalized FL methods in A.4.

**Computational cost.** We compare the total local time required by each client throughout 51 rounds and the final accuracy in Table 4. Note that we have include the generation time (which takes only 0.81 seconds for each round) required for FedCOG for fair comparison. 1) Results show that FedCOG introduces moderate training and generating time while achieving the highest accuracy. 2) This is acceptable as communication and privacy are more important in FL (Kairouz et al., 2019) while FedCOG makes no compromise on these two aspects, and the utility is improved with evident gain. Specifically, compared with FedProx and MOON, FedCOG achieves much higher performance with the least computation time.

Table 4: Local time of clients and test accuracy of global model. FedCOG introduces minor training cost while achieves the highest performance.

| Method | Local Time (s) | Acc (%) |
|---|---|---|
| FedAvg | **983** ±100 | 63.34 |
| FedProx | 1051 ± 87 | 64.57 |
| MOON | 1276 ± 147 | 64.44 |
| FedCOG | 1001 ± 98 | **65.59** |

**Applicability on different heterogeneity levels.** Here, we tune the argument $\beta \in \{0.1, 0.5, 1.0, 5.0\}$ for Dirichlet distribution in NIID-1 and compare with two representative baselines in Figure 3(a). Note that a smaller $\beta$ denotes a more heterogeneous setting. From the figure, we see that 1) generally, FL methods have worse performance under the more heterogeneous scenarios, indicating that data heterogeneity is a critical issue that limits FL performance. 2) FedCOG consistently performs the best across different heterogeneity levels, indicating its effectiveness on tackling data heterogeneity.

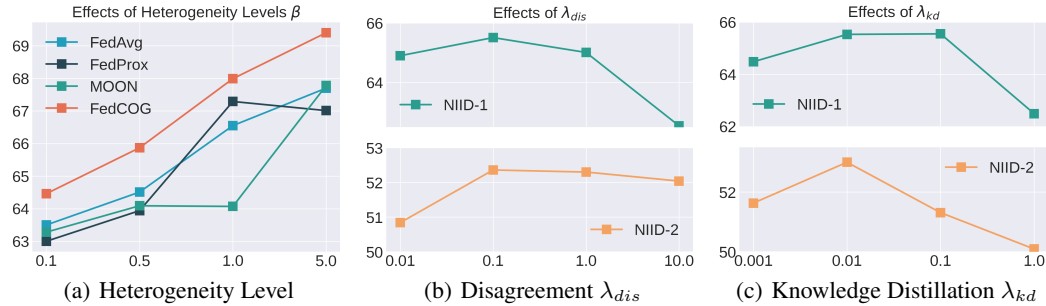

Figure 3: Ablation study. (a) FedCOG is effective across different heterogeneity levels. (b) Effects of disagreement in data generation. (c) Effects of knowledge distillation in model training.

## 5.5 ABLATION STUDY

To evaluate the impact of each module on the final performance of global model, we conduct experiments on CIFAR-10 for ablation study. See effects of FL arguments in Section A.6.

**Complementary data from noise or generation.** To validate the effectiveness of task-specific data generation from global model and local model, we also run experiments on pure Gaussian noise. From ① & ④ in Table 5, we see that data generation from global model and local model leads to significantly better performance than Gaussian noise, indicating the effectiveness of data generation through optimizing the inputs.

Table 5: Ablation study of FedCOG. ①: Complementary data is from Gaussian noise. ②: Data is generated from global model. ③: Hard-label supervision on generated data. ④: Full version of FedCOG.

| No. | Gen. Data Source | Super. | NIID-1 | NIID-2 |
|---|---|---|---|---|
| ① | Gaussian Noise | Soft | 60.98 | 49.67 |
| ② | Gen. from GM | Soft | 64.72 | 51.61 |
| ③ | Gen. from GM&LM | Hard | 63.59 | 51.64 |
| ④ | Gen. from GM&LM | Soft | **64.88** | **53.20** |

**Generating data from global model or global & local models.** By comparing ② with ④ in Table 5, we can see that generating data from merely global model achieves moderate performance while adding the disagreement term between global and local model contributes to better performance, indicating the effectiveness of disagreement loss in FedCOG. More detailed analysis is in Figure 3(b).

**Supervision in hard- or soft-label format.** By comparing ③ and ④ in Table 5, we see that soft-label format supervision leads to better performance, indicating the effectiveness of knowledge-distillation-based model training in FedCOG. More detailed analysis is in Figure 3(c).

**Effects of disagreement in generation and knowledge distillation in model training.** Under two heterogeneous settings on CIFAR-10, we tune $\lambda_{dis} \in \{0.01, 0.1, 1.0, 10.0\}$ and $\lambda_{kd} \in \{0.001, 0.01, 0.1, 1.0\}$, showing results in Figure 3(b) and 3(c), respectively. From the figure, we see that generally $\lambda_{dis} = 0.1$ and $\lambda_{kd} = 0.01$ can lead to better performance for both settings.

## 6 CONCLUSION AND FUTURE WORKS

In this paper, we seek to tackle the issue of data heterogeneity in FL from the novel perspective of modifying local dataset. To achieve this, we propose a novel FL algorithm, federated learning with consensus-oriented generation (FedCOG), to mitigate the heterogeneity level. FedCOG consists of two key components, complementary data generation to reduce heterogeneity level and knowledge-distillation-based model training to mitigate the effects of heterogeneity. FedCOG is plug-and-play in most existing FL methods, is compatible with standard FL protocol such as Secure Aggregation, and makes no compromise on communication cost and privacy while improves utility. Extensive experiments on classical and real-world FL datasets show FedCOG consistently outperforms state-of-the-art methods. We believe that FedCOG can inspire more future subsequent works along this line via adding regularization terms during generation or introducing advanced generative models.

ACKNOWLEDGEMENT

This research is supported by the National Key R&D Program of China under Grant 2021ZD0112801, NSFC under Grant 62171276 and the Science and Technology Commission of Shanghai Municipal under Grant 21511100900 and 22DZ2229005.

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

## A APPENDIX

### A.1 DATA GENERATION

#### A.1.1 DETAILS OF DESIGNING TARGET LABELS FOR GENERATING DATA

One basic generating strategy is uniformly generating data for all possible target labels. Taking CIFAR-10 as a concrete task example, the target label list can be $[\underbrace{0, ..., 0}_{n}, \underbrace{1, ..., 1}_{n}, ..., \underbrace{9, ..., 9}_{n}]$, resulting in $10 \times n$ samples to generate.

Though, there can also be more specific designs for particular tasks such that the generated dataset can better complement the original heterogeneous dataset. Here, we take a 5-way classification task as an example, where the ultimate objective is training a global model that works well fairly across categories while different clients have different preference on categories. Suppose the categorical distribution of client $k$ is $\boldsymbol{d}_k = [500, 0, 400, 200, 400]$, where the $i$-th element $\boldsymbol{d}_{k,i}$ denotes the number of samples that belongs to category $i$. Comparing with generating equal number of samples across categories, a more reasonable and targeted strategy is to generate more samples for those categories with fewer samples, better complementing the original dataset and empowering the client to perform better on those minor categories.

Specifically, in order to make the complimented dataset more uniform, we generate data according to the following complementary categorical data distribution $\hat{\boldsymbol{d}}_k = max(\boldsymbol{d}_k) - \boldsymbol{d}_k$, that is, $\hat{\boldsymbol{d}}_k = 500 - \boldsymbol{d}_k = [0, 500, 100, 300, 100]$. Then, given the budget of generating samples $N$, we will allocate $N \times \frac{\hat{\boldsymbol{d}}_{k,i}}{|\hat{\boldsymbol{d}}_k|}$ label $i$'s in the target label list, e.g., generating $N \times \frac{500}{1000}$ samples for category 1.

#### A.1.2 APPLICABILITY TO OTHER MODALITIES

Though we primarily focus on vision tasks to verify our idea of tackling data heterogeneity from data perspective, our method can also extend to other input modalities. Here, we elaborate this point on two types of modalities.

(1) For modalities such as continuous signals and images where the raw inputs can be optimized by gradient-based optimization, the pipeline is exactly the same as illustrated in this paper, since we can optimize the inputs in an end-to-end manner.

(2) For modalities such as text, where the raw inputs are harder to be directly optimized, it can be achieved with a slight difference. Since the raw inputs can not be optimized via end-to-end gradient-based optimization, we do not optimize/generate raw inputs (which is discrete) but optimize/generate the intermediate embeddings (which is continuous). For example, suppose the overall pipeline of the model is: [a] discrete inputs are transformed into continuous embeddings after going through tokenizing and embedding layers, [b] continuous embeddings go through the downstream task model, [c] the model gives outputs. Then, our data generation process is generating the continuous embeddings at step [b], which are regarded as consensus data.

### A.2 THEORETICAL CONVERGENCE ANALYSIS

For simplicity, assume that all the clients share the same generated consensus data (this can be achieved by generating data based on only global model and sharing the same random seed). We denote the global objective function as

$$\Phi(\boldsymbol{\theta}; \mathcal{D}) = \sum_{k=1}^{K} p_k \Phi_k(\boldsymbol{\theta}; \mathcal{D}) = \sum_{k=1}^{K} p_k \left[ F_k(\boldsymbol{\theta}) + Q_k(\boldsymbol{\theta}; \mathcal{D}) \right], \tag{4}$$

where $F_k$ is task loss, $Q_k$ is KD loss on generated dataset $\mathcal{D}$, and $\sum_{k=1}^{K} p_k = 1$. Then, we show four conventional assumptions in FL theoretical literature Wang et al. (2020b).

**Assumption A.1** (Smoothness). Each loss function $\Phi_k(\boldsymbol{\theta}; \mathcal{D})$ is Lipschitz-smooth.

**Assumption A.2** (Bounded Scalar). $\Phi_k(\boldsymbol{\theta}; \mathcal{D})$ is bounded below by $\Phi_{inf}$.

**Assumption A.3** (Unbiased Gradient and Bounded Variance). For each client, the stochastic gradient is unbiased: $\mathbb{E}_\xi[g_k(\boldsymbol{\theta}|\xi)] = \nabla\Phi_k(\boldsymbol{\theta};\mathcal{D})$, and has bounded variance: $\mathbb{E}_\xi[||g_k(\boldsymbol{\theta}|\xi) - \nabla\Phi_k(\boldsymbol{\theta};\mathcal{D})||^2] \leq \sigma^2$.

**Assumption A.4** (Bounded Dissimilarity). For any set of weights $\{p_k \geq 0\}_{k=1}^K$ subject to $\sum_{k=1}^K p_k = 1$, there exists constants $\beta^2 \geq 1$ and $\kappa^2 \geq 0$ such that $\sum_{k=1}^K p_k||\nabla\Phi_k(\boldsymbol{\theta};\mathcal{D})||^2 \leq \beta^2||\nabla\Phi(\boldsymbol{\theta};\mathcal{D})||^2 + \kappa^2$.

Additionally, we show a lemma that is critical for the derivation of our theoretical analysis.

**Lemma A.5** (Non-Increasing Global Loss).

$$\Phi(\boldsymbol{\theta}^{(t+1)};\mathcal{D}^{(t+1)}) \leq \Phi(\boldsymbol{\theta}^{(t+1)};\mathcal{D}^{(t)}). \tag{5}$$

*Proof.* We first expand both two sides of equation 5.

$$\Phi(\boldsymbol{\theta}^{(t+1)};\mathcal{D}^{(t+1)}) = F(\boldsymbol{\theta}^{(t+1)}) + Q(\boldsymbol{\theta}^{(t+1)};\mathcal{D}^{(t+1)}) = F(\boldsymbol{\theta}^{(t+1)}) + KL(\boldsymbol{\theta}^{(t+1)}||\boldsymbol{\theta}^{(t+1*)},\mathcal{D}^{(t+1)}), \tag{6}$$

$$\Phi(\boldsymbol{\theta}^{(t+1)};\mathcal{D}^{(t)}) = F(\boldsymbol{\theta}^{(t+1)}) + Q(\boldsymbol{\theta}^{(t+1)};\mathcal{D}^{(t)}) = F(\boldsymbol{\theta}^{(t+1)}) + KL(\boldsymbol{\theta}^{(t+1)}||\boldsymbol{\theta}^{(t*)},\mathcal{D}^{(t)}), \tag{7}$$

Since $KL(\boldsymbol{\theta}^{(t+1)}||\boldsymbol{\theta}^{(t+1*)},\mathcal{D}^{(t+1)}) = 0$ and $KL(\boldsymbol{\theta}^{(t+1)}||\boldsymbol{\theta}^{(t*)},\mathcal{D}^{(t)}) \geq 0$, we naturally have:

$$\Phi(\boldsymbol{\theta}^{(t+1)};\mathcal{D}^{(t+1)}) \leq \Phi(\boldsymbol{\theta}^{(t+1)};\mathcal{D}^{(t)}). \tag{8}$$

$\square$

Based on these assumptions and this lemma, we can finally prove the following theorem:

**Theorem A.6** (Optimization bound of the global objective function). *Under these assumptions, if we set $\eta L \leq \min\{\frac{1}{2\tau}, \frac{1}{\sqrt{2\tau(\tau-1)(2\beta^2+1)}}\}$, the optimization error will be bounded as follows:*

$$\min_t \mathbb{E}||\nabla\Phi(\boldsymbol{\theta}^{(t)};\mathcal{D}^{(t)})||^2$$
$$\leq \frac{4[\Phi(\boldsymbol{\theta}^{(0)};\mathcal{D}^{(0)}) - \Phi_{inf}]}{\tau\eta T} + 4\eta L\sigma^2 \sum_{k=1}^K p_k^2 + 3(\tau-1)\eta^2\sigma^2 L^2 + 6\tau(\tau-1)\eta^2 L^2\kappa^2, \tag{9}$$

*where $\eta$ is learning rate for local model training and $\tau$ is the number of local model updates.*

Theorem A.6 establishes that when the number of communication rounds, $T$, tends towards infinity ($T \rightarrow \infty$), the expectation of the optimization error remains bounded by a constant number for fixed $\eta$; see detailed proof in Appendix A.2.2. Further, based on Theorem A.6, we can deduce the following corollary when a suitable learning rate, denoted as $\eta$, is established:

**Corollary A.7** (Convergence of the global objective function). *By setting $\eta = \frac{1}{\sqrt{\tau T}}$, our method can converge to a stationary point given an infinite number of communication rounds $T$. Specifically, the bound could be reformulated as follows:*

$$\min_t \mathbb{E}||\nabla\Phi(\boldsymbol{\theta}^{(t)};\mathcal{D}^{(t)})||^2$$
$$\leq \frac{4[\Phi(\boldsymbol{\theta}^{(0)};\mathcal{D}^{(0)}) - \Phi_{inf}] + 4L\sigma^2 \sum_{k=1}^K p_k^2}{\sqrt{\tau T}} + \frac{3(\tau-1)\sigma^2 L^2}{\tau T} + \frac{6(\tau-1)L^2\kappa^2}{T} \tag{10}$$
$$= \mathcal{O}(\frac{1}{\sqrt{\tau T}}) + \mathcal{O}(\frac{1}{T}) + \mathcal{O}(\frac{\tau}{T}). \tag{11}$$

This corollary indicates that as $T \rightarrow \infty$, the error's upper bound approaches 0. Also, given a finite $T$, there exists a best $\tau$ that minimizes the error's upper bound. These analyses show that our method can achieve the same convergence rate as most methods, such as FedAvg Li et al. (2019); Wang et al. (2020b). Please see detailed proof in Appendix A.2.2.

### A.2.1 PRELIMINARIES

For ease of writing, we use $g_k(\boldsymbol{\theta})$ to denote mini-batch gradient $g_k(\boldsymbol{\theta}|\xi)$ and $\nabla\Phi_k(\boldsymbol{\theta})$ to denote full-batch gradient. We further define the following two notions:

$$\text{Averaged Mini-batch Gradient:} \quad \mathbf{d}_k^{(t)} = \frac{1}{\tau}\sum_{r=0}^{\tau-1} g_k(\boldsymbol{\theta}_k^{(t,r)}), \tag{12}$$

$$\text{Averaged Full-batch Gradient:} \quad \mathbf{h}_k^{(t)} = \frac{1}{\tau}\sum_{r=0}^{\tau-1} \nabla\Phi_k(\boldsymbol{\theta}_k^{(t,r)}). \tag{13}$$

Then, the update of the global model between two rounds is as follows:

$$\boldsymbol{\theta}^{(t+1)} - \boldsymbol{\theta}^{(t)} = -\tau\eta\sum_{k=1}^{K} p_k\mathbf{d}_k^{(t)}. \tag{14}$$

### A.2.2 PROOF OF THEOREM

According to the Lipschitz-smooth property in Assumption A.1, we have its equivalent form Bottou et al. (2018):

$$\mathbb{E}\left[\Phi(\boldsymbol{\theta}^{(t+1)};\mathcal{D}^{(t)})\right] - \Phi(\boldsymbol{\theta}^{(t)};\mathcal{D}^{(t)})$$

$$\leq \mathbb{E}\left[\left\langle\nabla\Phi(\boldsymbol{\theta}^{(t)};\mathcal{D}^{(t)}), \boldsymbol{\theta}^{(t+1)} - \boldsymbol{\theta}^{(t)}\right\rangle\right] - \frac{L}{2}\mathbb{E}\left[\left\|\boldsymbol{\theta}^{(t+1)} - \boldsymbol{\theta}^{(t)}\right\|^2\right] \tag{15}$$

$$= \underbrace{-\tau\eta\,\mathbb{E}\left[\left\langle\nabla\Phi(\boldsymbol{\theta}^{(t)};\mathcal{D}^{(t)}), \sum_{k=1}^{K} p_k\mathbf{d}_k^{(t)}\right\rangle\right]}_{T_1} + \underbrace{\frac{L\tau^2\eta^2}{2}\mathbb{E}\left[\left\|\sum_{k=1}^{K} p_k\mathbf{d}_k^{(t)}\right\|^2\right]}_{T_2}, \tag{16}$$

where the expectation is taken over mini-batches $\xi_k^{(t,r)}, \forall k \in 1, 2, ..., K, r \in 0, 1, ..., \tau - 1$.

For ease of writing, we use $\Phi(\boldsymbol{\theta}^{(t)})$ and $\Phi_k(\boldsymbol{\theta}^{(t)})$ to denote $\Phi(\boldsymbol{\theta}^{(t)};\mathcal{D}^{(t)})$ and $\Phi_k(\boldsymbol{\theta}^{(t)};\mathcal{D}^{(t)})$ when it causes no ambiguity for the rest of paper.

**Bounding $T_1$ in (16):**

$$T_1 = \mathbb{E}\left[\left\langle\nabla\Phi(\boldsymbol{\theta}^{(t)}), \sum_{k=1}^{K} p_k(\mathbf{d}_k^{(t)} - \mathbf{h}_k^{(t)})\right\rangle\right] + \mathbb{E}\left[\left\langle\nabla\Phi(\boldsymbol{\theta}^{(t)}), \sum_{k=1}^{K} p_k\mathbf{h}_k^{(t)}\right\rangle\right] \tag{17}$$

$$= \mathbb{E}\left[\left\langle\nabla\Phi(\boldsymbol{\theta}^{(t)}), \sum_{k=1}^{K} p_k\mathbf{h}_k^{(t)}\right\rangle\right] \tag{18}$$

$$= \frac{1}{2}\left\|\nabla\Phi(\boldsymbol{\theta}^{(t)})\right\|^2 + \frac{1}{2}\mathbb{E}\left[\left\|\sum_{k=1}^{K} p_k\mathbf{h}_k^{(t)}\right\|^2\right] - \frac{1}{2}\mathbb{E}\left[\left\|\nabla\Phi(\boldsymbol{\theta}^{(t)}) - \sum_{k=1}^{K} p_k\mathbf{h}_k^{(t)}\right\|^2\right], \tag{19}$$

where (18) uses the unbiased gradient assumption in Assumption A.3, such that $\mathbb{E}[\mathbf{d}_k^{(t)} - \mathbf{h}_i^{(t)}] = \mathbf{h}_k^{(t)} - \mathbf{h}_i^{(t)} = \mathbf{0}$. (19) uses the fact that $2\langle a, b\rangle = \|a\|^2 + \|b\|^2 - \|a - b\|^2$.

**Bounding $T_2$ in (16):**

$$T_2 = \mathbb{E}\left[\left\|\sum_{k=1}^{K} p_k(\mathbf{d}_k^{(t)} - \mathbf{h}_k^{(t)}) + \sum_{k=1}^{K} p_k \mathbf{h}_k^{(t)}\right\|^2\right] \tag{20}$$

$$\leq 2\mathbb{E}\left[\left\|\sum_{k=1}^{K} p_k(\mathbf{d}_k^{(t)} - \mathbf{h}_k^{(t)})\right\|^2\right] + 2\mathbb{E}\left[\left\|\sum_{k=1}^{K} p_k \mathbf{h}_k^{(t)}\right\|^2\right] \tag{21}$$

$$= 2\sum_{k=1}^{K} p_k^2 \mathbb{E}\left[\left\|\mathbf{d}_k^{(t)} - \mathbf{h}_k^{(t)}\right\|^2\right] + 2\mathbb{E}\left[\left\|\sum_{k=1}^{K} p_k \mathbf{h}_k^{(t)}\right\|^2\right] \tag{22}$$

$$= \frac{2}{\tau^2}\sum_{k=1}^{K} p_k^2 \mathbb{E}\left[\left\|\sum_{r=0}^{\tau-1}(g_k(\boldsymbol{\theta}_k^{(t,r)}) - \nabla\Phi_k(\boldsymbol{\theta}_k^{(t,r)}))\right\|^2\right] + 2\mathbb{E}\left[\left\|\sum_{k=1}^{K} p_k \mathbf{h}_k^{(t)}\right\|^2\right] \tag{23}$$

$$= \frac{2}{\tau^2}\sum_{k=1}^{K} p_k^2 \sum_{r=0}^{\tau-1} \mathbb{E}\left[\left\|g_k(\boldsymbol{\theta}_k^{(t,r)}) - \nabla\Phi_k(\boldsymbol{\theta}_k^{(t,r)})\right\|^2\right] + 2\mathbb{E}\left[\left\|\sum_{k=1}^{K} p_k \mathbf{h}_k^{(t)}\right\|^2\right] \tag{24}$$

$$\leq \frac{2\sigma^2}{\tau}\sum_{k=1}^{K} p_k^2 + 2\mathbb{E}\left[\left\|\sum_{k=1}^{K} p_k \mathbf{h}_k^{(t)}\right\|^2\right] \tag{25}$$

where (21) uses $\|a+b\|^2 \leq 2\|a\|^2 + 2\|b\|^2$, (22) uses the fact that clients are independent to each other so that $\mathbb{E}\left\langle \mathbf{d}_k^{(t)} - \mathbf{h}_k^{(t)}, \mathbf{d}_n^{(t)} - \mathbf{h}_n^{(t)}\right\rangle = 0, \forall k \neq n$. (24) uses Lemma 2 in Wang et al. (2020b) and (25) uses bounded variance assumption in Assumption A.3.

Plug (19) and (25) back into (16), we have

$$\mathbb{E}\left[\Phi(\boldsymbol{\theta}^{(t+1)}; \mathcal{D}^{(t)})\right] - \Phi(\boldsymbol{\theta}^{(t)}; \mathcal{D}^{(t)})$$

$$\leq -\frac{\tau\eta}{2}\left\|\nabla\Phi(\boldsymbol{\theta}^{(t)})\right\|^2 - \frac{\tau\eta}{2}(1 - 2\tau\eta L)\mathbb{E}\left[\left\|\sum_{k=1}^{K} p_k \mathbf{h}_k^{(t)}\right\|^2\right]$$

$$+ L\tau\eta^2\sigma^2 \sum_{k=1}^{K} p_k^2 + \frac{\tau\eta}{2}\mathbb{E}\left[\left\|\nabla\Phi(\boldsymbol{\theta}^{(t)}) - \sum_{k=1}^{K} p_k \mathbf{h}_k^{(t)}\right\|^2\right]. \tag{26}$$

When $1 - 2\tau\eta L \geq 0$, we have

$$\mathbb{E}\left[\Phi(\boldsymbol{\theta}^{(t+1)}; \mathcal{D}^{(t)})\right] - \Phi(\boldsymbol{\theta}^{(t)}; \mathcal{D}^{(t)})$$

$$\leq -\frac{\tau\eta}{2}\left\|\nabla\Phi(\boldsymbol{\theta}^{(t)})\right\|^2 + L\tau\eta^2\sigma^2 \sum_{k=1}^{K} p_k^2 + \frac{\tau\eta}{2}\mathbb{E}\left[\left\|\nabla\Phi(\boldsymbol{\theta}^{(t)}) - \sum_{k=1}^{K} p_k \mathbf{h}_k^{(t)}\right\|^2\right] \tag{27}$$

$$= -\frac{\tau\eta}{2}\left\|\nabla\Phi(\boldsymbol{\theta}^{(t)})\right\|^2 + L\tau\eta^2\sigma^2 \sum_{k=1}^{K} p_k^2 + \frac{\tau\eta}{2}\mathbb{E}\left[\left\|\sum_{k=1}^{K} p_k(\nabla\Phi_k(\boldsymbol{\theta}^{(t)}) - \mathbf{h}_k^{(t)})\right\|^2\right] \tag{28}$$

$$\leq -\frac{\tau\eta}{2}\left\|\nabla\Phi(\boldsymbol{\theta}^{(t)})\right\|^2 + L\tau\eta^2\sigma^2 \sum_{k=1}^{K} p_k^2 + \frac{\tau\eta}{2}\sum_{k=1}^{K} p_k \underbrace{\mathbb{E}\left[\left\|\nabla\Phi_k(\boldsymbol{\theta}^{(t)}) - \mathbf{h}_k^{(t)}\right\|^2\right]}_{T_3}, \tag{29}$$

where (29) uses Jensen's Inequality $\left\|\sum_{k=1}^{K} p_k x_k\right\|^2 \leq \sum_{k=1}^{K} p_k \|x_k\|^2$.

**Bounding $T_3$ in (29):**

$$\mathbb{E}\left[\left\|\nabla\Phi_k(\boldsymbol{\theta}^{(t)}) - \mathbf{h}_k^{(t)}\right\|^2\right] = \mathbb{E}\left[\left\|\nabla\Phi_k(\boldsymbol{\theta}^{(t)}) - \frac{1}{\tau}\sum_{r=0}^{\tau-1}\nabla\Phi_k(\boldsymbol{\theta}_k^{(t,r)})\right\|^2\right] \tag{30}$$

$$= \mathbb{E}\left[\left\|\frac{1}{\tau}\sum_{r=0}^{\tau-1}(\nabla\Phi_k(\boldsymbol{\theta}^{(t)}) - \nabla\Phi_k(\boldsymbol{\theta}_k^{(t,r)}))\right\|^2\right] \tag{31}$$

$$\leq \frac{1}{\tau}\sum_{r=0}^{\tau-1}\mathbb{E}\left[\left\|\nabla\Phi_k(\boldsymbol{\theta}^{(t)}) - \nabla\Phi_k(\boldsymbol{\theta}_k^{(t,r)})\right\|^2\right] \tag{32}$$

$$\leq \frac{L^2}{\tau}\sum_{r=0}^{\tau-1}\underbrace{\mathbb{E}\left[\left\|\boldsymbol{\theta}^{(t)} - \boldsymbol{\theta}_k^{(t,r)}\right\|^2\right]}_{T_4}, \tag{33}$$

where (32) uses Jensen's Inequality and (33) follows Lipschitz-smooth property.

**Bounding $T_4$ in (33):**

$$\mathbb{E}\left[\left\|\boldsymbol{\theta}^{(t)} - \boldsymbol{\theta}_k^{(t,r)}\right\|^2\right] = \eta^2\mathbb{E}\left[\left\|\sum_{s=0}^{r-1}g_k(\boldsymbol{\theta}_k^{(t,s)})\right\|^2\right] \tag{34}$$

$$\leq 2\eta^2\mathbb{E}\left[\left\|\sum_{s=0}^{r-1}\left(g_k(\boldsymbol{\theta}_k^{(t,s)}) - \nabla\Phi_k(\boldsymbol{\theta}_k^{(t,s)})\right)\right\|^2\right] + 2\eta^2\mathbb{E}\left[\left\|\sum_{s=0}^{r-1}\nabla\Phi_k(\boldsymbol{\theta}_k^{(t,s)})\right\|^2\right] \tag{35}$$

$$= 2\eta^2\sum_{s=0}^{r-1}\mathbb{E}\left[\left\|g_k(\boldsymbol{\theta}_k^{(t,s)}) - \nabla\Phi_k(\boldsymbol{\theta}_k^{(t,s)})\right\|^2\right] + 2\eta^2\mathbb{E}\left[\left\|\sum_{s=0}^{r-1}\nabla\Phi_k(\boldsymbol{\theta}_k^{(t,s)})\right\|^2\right] \tag{36}$$

$$\leq 2r\eta^2\sigma^2 + 2\eta^2\mathbb{E}\left[\left\|r\sum_{s=0}^{r-1}\frac{1}{r}\nabla\Phi_k(\boldsymbol{\theta}_k^{(t,s)})\right\|^2\right] \tag{37}$$

$$\leq 2r\eta^2\sigma^2 + 2r\eta^2\sum_{s=0}^{r-1}\mathbb{E}\left[\left\|\nabla\Phi_k(\boldsymbol{\theta}_k^{(t,s)})\right\|^2\right] \tag{38}$$

$$\leq 2r\eta^2\sigma^2 + 2r\eta^2\sum_{s=0}^{\tau-1}\mathbb{E}\left[\left\|\nabla\Phi_k(\boldsymbol{\theta}_k^{(t,s)})\right\|^2\right] \tag{39}$$

where (35) uses $\|a + b\|^2 \leq 2\|a\|^2 + 2\|b\|^2$, (36) uses Lemma 2 in Wang et al. (2020b), (37) uses the bounded variance assumption in Assumption A.3, (38) uses Jensen's Inequality.

Plug (39) back into (33) and use this equation $\sum_{r=0}^{\tau-1} r = \frac{\tau(\tau-1)}{2}$, we have

$$\mathbb{E}\left[\left\|\nabla\Phi_k(\boldsymbol{\theta}^{(t)}) - \mathbf{h}_k^{(t)}\right\|^2\right]$$

$$\leq \frac{L^2}{\tau}\sum_{r=0}^{\tau-1}\mathbb{E}\left[\left\|\boldsymbol{\theta}^{(t)} - \boldsymbol{\theta}_k^{(t,r)}\right\|^2\right] \tag{40}$$

$$\leq (\tau-1)L^2\eta^2\sigma^2 + (\tau-1)L^2\eta^2\sum_{s=0}^{\tau-1}\underbrace{\mathbb{E}\left[\left\|\nabla\Phi_k(\boldsymbol{\theta}_k^{(t,s)})\right\|^2\right]}_{T_5}, \tag{41}$$

where $T_5$ in (41) can be further bounded.

**Bounding $T_5$ in (41):**

$$\mathbb{E}\left[\left\|\nabla\Phi_k(\boldsymbol{\theta}_k^{(t,s)})\right\|^2\right]$$

$$\leq 2\mathbb{E}\left[\left\|\nabla\Phi_k(\boldsymbol{\theta}_k^{(t,s)}) - \nabla\Phi_k(\boldsymbol{\theta}^{(t)})\right\|^2\right] + 2\mathbb{E}\left[\left\|\nabla\Phi_k(\boldsymbol{\theta}^{(t)})\right\|^2\right] \tag{42}$$

$$\leq 2L^2\mathbb{E}\left[\left\|\boldsymbol{\theta}^{(t)} - \boldsymbol{\theta}_k^{(t,s)}\right\|^2\right] + 2\mathbb{E}\left[\left\|\nabla\Phi_k(\boldsymbol{\theta}^{(t)})\right\|^2\right], \tag{43}$$

where (42) uses $\|a+b\|^2 \leq 2\|a\|^2 + 2\|b\|^2$, (43) uses Lipschitz-smooth property. Plug (43) back to (41), we have

$$\frac{L^2}{\tau}\sum_{r=0}^{\tau-1}\mathbb{E}\left[\left\|\boldsymbol{\theta}^{(t)} - \boldsymbol{\theta}_k^{(t,r)}\right\|^2\right]$$

$$\leq (\tau-1)L^2\eta^2\sigma^2 + 2(\tau-1)\eta^2L^4\sum_{s=0}^{\tau-1}\mathbb{E}\left[\left\|\boldsymbol{\theta}_k^{(t,0)} - \boldsymbol{\theta}^{(t,s)}\right\|^2\right] + 2(\tau-1)\eta^2L^2\sum_{s=0}^{\tau-1}\mathbb{E}\left[\left\|\nabla\Phi_k(\boldsymbol{\theta}^{(t)})\right\|^2\right] \tag{44}$$

After rearranging, we have

$$\mathbb{E}\left[\left\|\nabla\Phi_k(\boldsymbol{\theta}^{(t)}) - \mathbf{h}_k^{(t)}\right\|^2\right]$$

$$\leq \frac{L^2}{\tau}\sum_{r=0}^{\tau-1}\mathbb{E}\left[\left\|\boldsymbol{\theta}^{(t)} - \boldsymbol{\theta}_k^{(t,r)}\right\|^2\right] \tag{45}$$

$$\leq \frac{(\tau-1)\eta^2\sigma^2L^2}{1-2\tau(\tau-1)\eta^2L^2} + \frac{2\tau(\tau-1)\eta^2L^2}{1-2\tau(\tau-1)\eta^2L^2}\mathbb{E}\left[\left\|\nabla\Phi_k(\boldsymbol{\theta}^{(t)})\right\|^2\right] \tag{46}$$

$$= \frac{(\tau-1)\eta^2\sigma^2L^2}{1-A} + \frac{A}{1-A}\mathbb{E}\left[\left\|\nabla\Phi_k(\boldsymbol{\theta}^{(t)})\right\|^2\right], \tag{47}$$

where we define $A = 2\tau(\tau-1)\eta^2L^2 < 1$. Then

$$\frac{\tau\eta}{2}\sum_{k=1}^{K}p_k\mathbb{E}\left[\left\|\nabla\Phi_k(\boldsymbol{\theta}^{(t)}) - \mathbf{h}_k^{(t)}\right\|^2\right]$$

$$\leq \frac{\tau\eta}{2}\sum_{k=1}^{K}\left\{p_k\left[\frac{(\tau-1)\eta^2\sigma^2L^2}{1-A} + \frac{A}{1-A}\mathbb{E}\left[\left\|\nabla\Phi_k(\boldsymbol{\theta}^{(t)})\right\|^2\right]\right]\right\} \tag{48}$$

$$\leq \frac{\tau(\tau-1)\sigma^2L^2\eta^3}{2(1-A)} + \frac{A\tau\eta\beta^2}{2(1-A)}\mathbb{E}\left[\left\|\nabla\Phi(\boldsymbol{\theta}^{(t)})\right\|^2\right] + \frac{A\tau\eta\kappa^2}{2(1-A)}, \tag{49}$$

where (49) follows bounded dissimilarity assumption in Assumption A.4. Plug (49) back to (29), we have

$$\mathbb{E}\left[\Phi(\boldsymbol{\theta}^{(t+1)};\mathcal{D}^{(t)})\right] - \Phi(\boldsymbol{\theta}^{(t)};\mathcal{D}^{(t)})$$

$$\leq -\frac{\tau\eta}{2}\left\|\nabla\Phi(\boldsymbol{\theta}^{(t)};\mathcal{D}^{(t)})\right\|^2 + L\tau\eta^2\sigma^2\sum_{k=1}^{K}p_k^2$$

$$+ \frac{\tau(\tau-1)\sigma^2L^2\eta^3}{2(1-A)} + \frac{A\tau\eta\beta^2}{2(1-A)}\mathbb{E}\left[\left\|\nabla\Phi(\boldsymbol{\theta}^{(t)};\mathcal{D}^{(t)})\right\|^2\right] + \frac{A\tau\eta\kappa^2}{2(1-A)} \tag{50}$$

$$= -\frac{\tau\eta}{2}\left(1 - \frac{A\beta^2}{1-A}\right)\left\|\nabla\Phi(\boldsymbol{\theta}^{(t)};\mathcal{D}^{(t)})\right\|^2 + L\tau\eta^2\sigma^2\sum_{k=1}^{K}p_k^2 + \frac{\tau(\tau-1)\sigma^2L^2\eta^3}{2(1-A)} + \frac{A\tau\eta\kappa^2}{2(1-A)}. \tag{51}$$

If $\frac{A\beta^2}{1-A} \le \frac{1}{2}$, then $\frac{1}{1-A} \le 1 + \frac{1}{2\beta^2}$ and we have

$$\mathbb{E}\left[\Phi(\boldsymbol{\theta}^{(t+1)}; \mathcal{D}^{(t)})\right] - \Phi(\boldsymbol{\theta}^{(t)}; \mathcal{D}^{(t)})$$

$$\le -\frac{\tau\eta}{4}\left\|\nabla\Phi(\boldsymbol{\theta}^{(t)}; \mathcal{D}^{(t)})\right\|^2 + L\tau\eta^2\sigma^2\sum_{k=1}^{K}p_k^2 + \frac{\tau(\tau-1)\sigma^2 L^2\eta^3}{2}(1+\frac{1}{2\beta^2}) + \frac{A\tau\eta\kappa^2}{2}(1+\frac{1}{2\beta^2})$$
(52)

$$\le -\frac{\tau\eta}{4}\left\|\nabla\Phi(\boldsymbol{\theta}^{(t)}; \mathcal{D}^{(t)})\right\|^2 + L\tau\eta^2\sigma^2\sum_{k=1}^{K}p_k^2 + \frac{3}{4}\tau(\tau-1)\sigma^2 L^2\eta^3 + \frac{3}{4}A\tau\eta\kappa^2,$$
(53)

where (53) follows $\beta \ge 1$ in Assumption A.4. Then

$$\mathbb{E}\left[\Phi(\boldsymbol{\theta}^{(t+1)}; \mathcal{D}^{(t+1)})\right] - \Phi(\boldsymbol{\theta}^{(t)}; \mathcal{D}^{(t)})$$

$$\le \mathbb{E}\left[\Phi(\boldsymbol{\theta}^{(t+1)}; \mathcal{D}^{(t)})\right] - \Phi(\boldsymbol{\theta}^{(t)}; \mathcal{D}^{(t)})$$
(54)

$$\le -\frac{\tau\eta}{4}\left\|\nabla\Phi(\boldsymbol{\theta}^{(t)}; \mathcal{D}^{(t)})\right\|^2 + L\tau\eta^2\sigma^2\sum_{k=1}^{K}p_k^2 + \frac{3}{4}\tau(\tau-1)\sigma^2 L^2\eta^3 + \frac{3}{4}A\tau\eta\kappa^2,$$
(55)

where (54) follows our key lemma in Lemma A.5.

Finally, by taking the average expectation across all rounds, we finish the proof of our Theorem A.6:

$$\min_t \mathbb{E}\left\|\nabla\Phi(\boldsymbol{\theta}^{(t)}; \mathcal{D}^{(t)})\right\|^2 \le \frac{1}{T}\sum_{t=0}^{T-1}\mathbb{E}\left\|\nabla\Phi(\boldsymbol{\theta}^{(t)}; \mathcal{D}^{(t)})\right\|^2$$
(56)

$$\le \frac{4\left[\Phi(\boldsymbol{\theta}^{(0,0)}) - \Phi_{inf}\right]}{\tau\eta T} + 4\eta L\sigma^2\sum_{k=1}^{K}p_k^2 + 3(\tau-1)\eta^2\sigma^2 L^2 + 6\tau(\tau-1)\eta^2 L^2\kappa^2.$$
(57)

**Constraints on local learning rate.** Here, we summarize the constraints on local learning rate $\eta$:

$$1 - 2\tau\eta L \ge 0,$$
(58)

$$\frac{1}{1 - 2\tau(\tau-1)\eta^2 L^2} \le 1 + \frac{1}{2\beta^2},$$
(59)

that is,

$$\eta L \le \min\left\{\frac{1}{2\tau}, \frac{1}{\sqrt{2\tau(\tau-1)(2\beta^2+1)}}\right\}.$$
(60)

### A.2.3 COROLLARY

By setting $\eta = \frac{1}{\sqrt{\tau T}}$, the above bound can be written as

$$\min_t \mathbb{E}\left\|\nabla\Phi(\boldsymbol{\theta}^{(t)}; \mathcal{D}^{(t)})\right\|^2 \le \frac{4[\Phi(\boldsymbol{\theta}^{(0,0)}; \mathcal{D}^0) - \Phi_{inf}] + 4L^2\sigma^2\sum_{k=1}^{K}p_k^2}{\sqrt{\tau T}}$$

$$+ \frac{3(\tau-1)\sigma^2 L^2}{\tau T} + \frac{6\tau(\tau-1)L^2\kappa^2}{\tau T}$$
(61)

$$= \mathcal{O}(\frac{1}{\sqrt{\tau T}}) + \mathcal{O}(\frac{1}{T}) + \mathcal{O}(\frac{\tau}{T}).$$
(62)

### A.3 EXPERIMENTAL DETAILS

### A.3.1 IMPLEMENTATION DETAILS

**Datasets and Heterogeneity.** Overall, we consider three classical datasets and one real-world FL dataset.

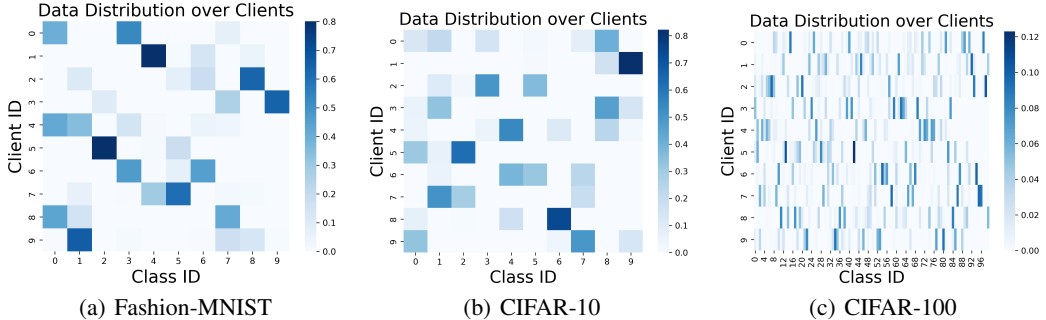

(a) Fashion-MNIST        (b) CIFAR-10        (c) CIFAR-100

Figure 4: Data distribution of NIID-1 for Fashion-MNIST, CIFAR-10 and CIFAR-100

1. Fashion-MNIST (Xiao et al., 2017) is a 10-category classification dataset that consists of 10 types of clothes;

2. CIFAR-10 and CIFAR-100 (Krizhevsky et al., 2009) are two classification datasets that consists of 10 and 100 types of objects, respectively;

3. FLAIR (Song et al., 2022) is a recently released real-world FL multi-label dataset, where each client is a user from Flickr with multiple images and corresponding tags. We adopt the task of predicting the existence of 17 categories of objects in each image.

For the three classical datasets, we consider two types of data heterogeneity, namely NIID-1 and NIID-2.

1. NIID-1 follows Dirichlet distribution $Dir_\beta$ (default $\beta = 0.1$), which is a common setting in FL (Yurochkin et al., 2019; Wang et al., 2020a;b; Li et al., 2021a; Shi et al., 2022; Fan et al., 2024). The argument $\beta$ controls the level of data heterogeneity, where a smaller $\beta$ corresponds to a more heterogeneous level and $\beta$ is set to 0.1 for default setting. The distribution for these datasets are shown in Figure. 4

2. NIID-2 makes each client hold data of only partial labels (McMahan et al., 2017; Li et al., 2020b) (2 for Fashion-MNIST and CIFAR-10, 10 for CIFAR-100).

For FLAIR (Song et al., 2022), the data heterogeneity is inherent as different local dataset is drawn from different user while different user has different preference and thus different data distribution.

**Models.** 1) For Fashion-MNIST and CIFAR-10, we use a simple convolutional neural network (CNN) that is frequently used in FL literature (Li et al., 2021a). The data flow in this model is: $5 \times 5$ convolution, max-pooling, $5 \times 5$ convolution, three fully-connected layer with hidden size of 120, 84 and 10 respectively. 2) For CIFAR-100, we use a slightly different simple CNN. The data flow in this model is: $3 \times 3$ convolution, max-pooling, $3 \times 3$ convolution, max-pooling, $3 \times 3$ convolution, two fully-connected layer with hidden size of 128 and 100 respectively. 3) For FLAIR, we use ResNet18 (He et al., 2016) from PyTorch API.

A.3.2    HYPER-PARAMETER-FREE KNOWLEDGE DISTILLATION FOR IMAGE CLASSIFICATION.

Following the example in A.1.1, for a classification task, we also propose a hyper-parameter-free knowledge distillation manner. Specifically, denote the number of samples from the original real dataset as $N_{real} = |\boldsymbol{d}_k|$ and the number of samples required to complement a balanced dataset $N_{gen} = |\hat{\boldsymbol{d}}_k|$, we set the coefficient of task-driven loss $\mathcal{L}_c$ as $\frac{N_{real}}{N_{real}+N_{gen}}$ and the coefficient of knowledge-distillation loss $\mathcal{L}_{kd}$ as $\frac{N_{gen}}{N_{real}+N_{gen}}$. In this case, samples from all categories are supervised with similar intensity, relieving the issue of imbalanced learning (Menon et al., 2021; Hong et al., 2023; Zhou et al., 2022; 2023) and contributing to enhanced overall performance.

**Baselines and hyper-parameters.** Here, we list the applied hyper-parameters for the compared baselines.

1. FedAvgM (Hsu et al., 2019): the momentum coefficient is set to $0.1$.

2. FedProx (Li et al., 2020b): the model regularization strength $\mu$ is set to $0.01$.

3. MOON (Li et al., 2021a): the feature regularization strength $\mu$ is set to $0.01$.

4. FedSAM (Qu et al., 2022): the constant to control the radius of the perturbation $\rho$ is set to $0.5$.

5. VHL (Tang et al., 2022): the coefficient of conditional distribution mismatch penalty $\lambda$ is set to $1.0$ and the number of epochs for feature alignment is set to $5$.

6. FedExP (Jhunjhunwala et al., 2023): the $\epsilon$ is set to $1e-3$. Note that in our experiments, the performance of FedExP is unstable. Through careful checking, we find that the experiments in the original paper of FedExP are conducted by setting local iteration $\tau = 20$. While in our paper, we are more interested in larger number of iterations (e.g., $\tau = 400$) which is common in FL. We conjecture that such large iteration number causes the model update more divergent and thus leads to unstable performance of FedExP.

7. FedDecorr (Shi et al., 2022): the feature regularization strength $\beta$ is set to $0.01$.

## A.4 Personalization analysis

**Methodology.** FedCOG provides a better global model, which further provides a better initialization for local, personalized models. Specifically, with data generation and knowledge distillation, FedCOG achieves a better-performed aggregated global model, which is subsequently used as initialization for local model training. Such a better-performed global model provides a better initial point for local model training and thus benefits the model training (improves the personalization). This phenomenon is actually common as FedAvg with Fine-tuning is generally a strong baseline in personalized FL (Collins et al., 2021; Ye et al., 2023a), which is also explained in (Collins et al., 2022).

**Experiments.** (1) In Figure 2(c), we have shown the advantages of FedCOG in the personalized setting by comparing it with two baselines in personalized FL: FedAvg/FedProx with Fine-tuning (local models before aggregation in FedAvg). Both baseline methods are strong and have been intensively verified in many personalized FL papers (Collins et al., 2021; Ye et al., 2023a; Collins et al., 2022). Through the figure, we see that FedCOG consistently achieves the highest accuracy across clients. (2) Besides, we conduct further evaluations on personalization by comparing with several representative personalized FL methods, namely, FedAvg with Finetuning, FedProx with Finetuning, FedRep (Collins et al., 2021), and Ditto (Li et al., 2021b). We conduct experiments on CIFAR-10 NIID-1 and the results are shown in Table 6. From the table, we see that FedCOG outperforms all these strong and representative baselines.

Table 6: Accuracy(%) comparisons with four representative personalized FL methods across CIFAR-10 dataset at NIID-1 heterogeneity level.

| Method | FedAvg+FT | FedProx+FT | FedRep | Ditto | **FedCOG+FT** |
|--------|-----------|------------|--------|-------|---------------|
| Acc % | 76.16 | 76.38 | 74.90 | 76.57 | **78.34** |

## A.5 Further Analysis of FedCOG

### A.5.1 Generated data analysis

Here, we provide analysis on the generated data.

**Visualization of generated data.** We visualize the generated data after various generation steps. CIFAR-10 consists of 32x32 colour images in 10 classes: airplane, automobile, bird, cat, deer, dog, frog, horse, ship and truck. From Figure 5, we see that 1) when the generation step is set to 0, the synthetic data produced is purely random noise. As the number of iterations increases, the generated images start to contain more specific patterns related to their corresponding classes. For instance, when generating images of ships and horses, the abstract shape of a hull and a horse are included in the output. 2) In this experiment, we validate the federated learning framework is privacy preserving,

Table 7: Effects of generation steps. A moderate size of generation step (e.g., 100) contributes to higher accuracy.

| Generation step | 0 | 50 | 100 | 500 | 1000 |
|---|---|---|---|---|---|
| Accuracy | 60.98 | 65.40 | 65.44 | 64.80 | 64.92 |

clients could not recover detailed private data from either client side or server side. The patterns in each image are highly abstract and contain certain category-specific features, but overall, we cannot invert a trained network to generate class-specific input images from random noise.

**Effect of generated data.** (1) The synthetic data is optimized to be correctly predicted by the global model, thus, it can be seen as a representative of the knowledge of global model. Then, during training, we use the synthetic data to distill knowledge from global model to local model, which basically forces the local model to not forget what the global model has already known. This would benefit the updating process of global model and will not be affected even if the synthetic data is not visually the same as real data. (2) We do not directly use the generated data to train the local model (i.e., using cross entropy loss to train local model without guidance from global model), which prevents the local model from overly fitting the generated data that could contain some unrelated properties.

**Number of generation steps.** Here we experiment on how the number of generation steps affects performance on CIFAR-10 and report the results in Table 7. From the table, we see that 1) generally, a moderate size (e.g., 100) of generation step contributes to better performance. 2) The performance is degraded when the generation step is too small or too large. This is reasonable as when the generation step is small, the image is similar to pure noise that may fail to well complement the original dataset; when the generation step is large, the optimizing process leads to over-fitting too many details and thus could even bring negative impacts.

**Effects of introducing generator.** In this experiment, we analyze the effects of introducing a learnable generator into the process of data generation. We insert a learnable convolutional neural network between the learnable noisy inputs and the taks-specific models. The introduced generator consists of a linear layer with hidden size 8192, a upsampling and three $3 \times 3$ convolution layers. After introducing the generator, we keep using the same hyper-parameters as before. We see that introducing a generator brings performance gain to NIID-2 setting (1.5% absolute accuracy improvement), but degrades performance to NIID-1 setting (0.93% absolute accuracy deterioration). This indicates that using the same set of hyper-parameters, FedCOG without generator and FedCOG with a learnable generator perform on par. Such results are reasonable since we do not carefully tune hyper-parameters and the generator is training from scratch. However, we believe that with careful hyper-parameter tuning and per-trained generator, the performance can be further improved, though, we leave this to future works.

### A.5.2 FEDCOG UNDER RESOURCE HETEROGENEITY

In practice, FL clients may vary in their available resources. Several methods can be employed to fit this scenario, including:

First, we can use a small batch size during data generation for resource-constrained clients, thereby mitigating memory constraints on these clients. By adjusting the batch size to match the limited memory capacity, we enable efficient data generation without necessitating any alterations to the underlying algorithm. An illustrative example from our CIFAR-100 experiment underscores the feasibility of this approach. Using a simple convolutional neural network, the required number of learnable parameters for local model training is 69380. While for instance, when employing a batch size of 1, the number of learnable parameters during local model training dramatically reduces to 3x32x32=3072, thereby rendering it practical to generate data for clients with restricted resources.

Second, another interesting angle is to use different model sizes for different clients. In a system where different clients have different computing resources, we can tailor model sizes to match individual capabilities (Diao et al., 2020; Horvath et al., 2021). By embracing this dynamic model-sizing

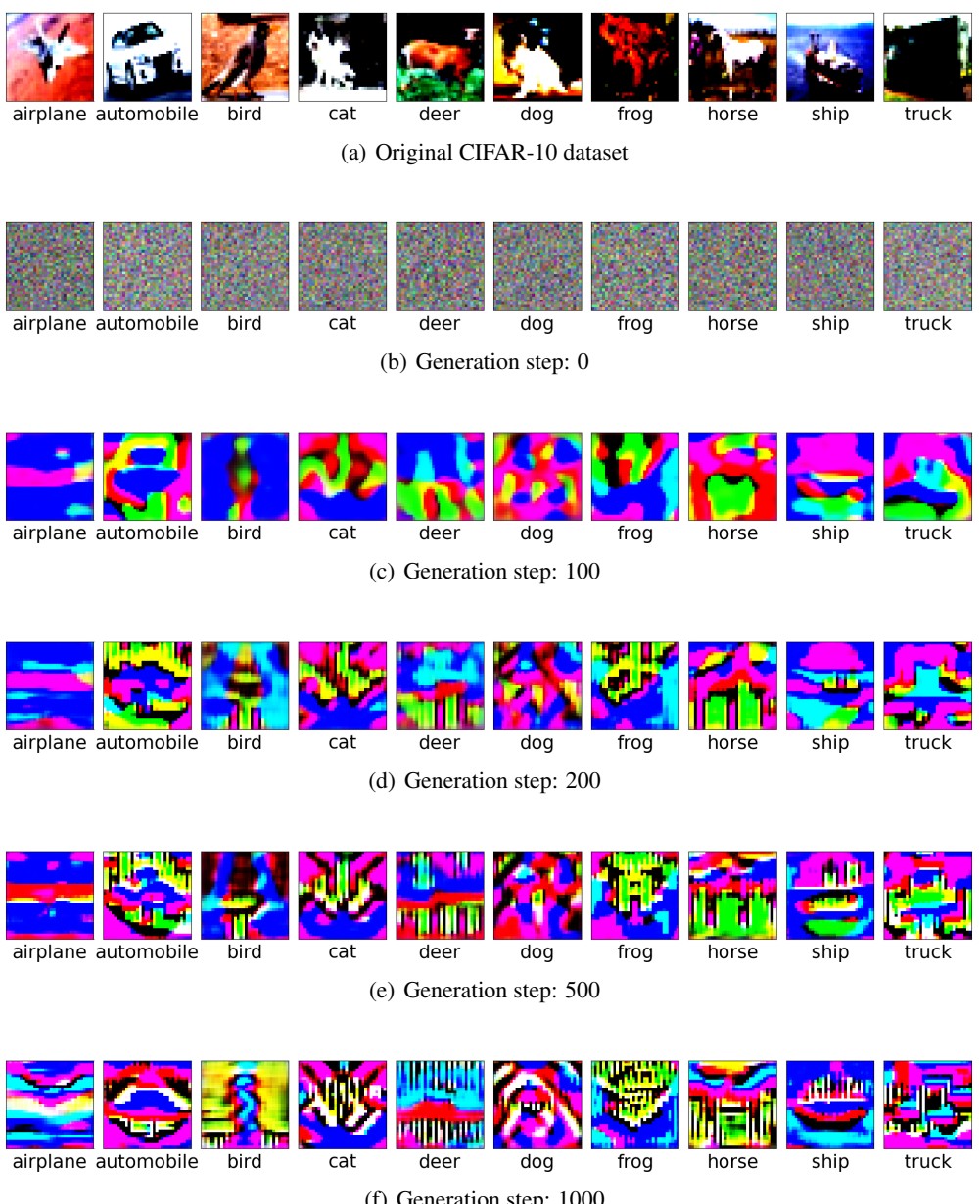

Figure 5: Visualization of generated data and CIFAR-10. The generated data contains some abstract patterns for the corresponding category, but will not reveal private details.

strategy, clients constrained by limited resources can harness models with reduced dimensions, such as employing sub-sampled CNNs with fewer channels (Diao et al., 2020).

Third, we introduce a new avenue for future exploration under our framework; that is, the adoption of variable generator sizes based on the clients' resource profiles. Specifically, resource-constrained clients solely focus on learning the input tensors (no generator, as done in our paper). While clients with more resources could learn to generate data with the help of a pre-trained generative model. We believe that this approach would be an interesting future direction.

### A.5.3  GENERATION TIME V.S. NUMBER OF GENERATED SAMPLES

The process of data generation in FedCOG is efficient as the only learnable parameters are the inputs (which is relatively small compared to model parameters) and the required optimization steps is only 100. To further verify this, we adjust the number of generated samples and record the required optimization time for each client at each round. We vary the number of generated samples in the range of [16, 32, 64, 128, 256, 512, 1024] and report the corresponding generation time in Table 8. From the table, we see that the generation process is quite efficient. Throughout the ppaer, we set the number of generated samples as 256, which costs only 0.81 seconds. As a reference, the conventional SGD-based local model training takes 14.81 seconds. Also, even when the number of generated samples is 1024, the required generation time is still only 1.27 seconds, indicating that our method is efficient.

Table 8: Required generation time under different numbers of generated samples.

| Num. | 16 | 32 | 64 | 128 | 256 | 512 | 1024 |
|------|------|------|------|------|------|------|------|
| Time | 0.72 | 0.78 | 0.79 | 0.80 | 0.81 | 0.83 | 1.27 |

### A.6  EFFECTS OF FL ARGUMENTS

In order to validate the robustness of our method for various federated learning settings, we conducted experiments using the CIFAR-10 dataset with NIID-1 distribution.

**Iteration number of local model training.** Here we evaluate the effect of the number of iteration for local training on CIFAR-10 with NIID-1 data distribution. The results are shown in Table 9. From the table, we see that 1) when $\tau$ is relatively small, the accuracy of all methods rise with the number of local iteration increasing; while when the number of local iteration gets too large (e.g., 800), the accuracy of all approaches drops due to the drift of local updates. 2) Nevertheless, our method FedCOG consistently outperforms the other methods in all scenarios.

Table 9: Effects of iteration number of local model training $\tau$. FedCOG consistently performs the best for different $\tau$.

| $\tau$ | 200 | 400 | 600 | 800 |
|--------|------|------|------|------|
| FedAvg | 61.46 | 64.68 | 63.45 | 62.39 |
| FedProx | 61.39 | 63.90 | 62.13 | 61.48 |
| MOON | 61.12 | 63.59 | 63.27 | 62.81 |
| **FedCOG** | **61.73** | **65.61** | **64.89** | **64.98** |

**Partial client participation scenarios.** For this experiment, we consider $K = 50$ clients in total and sample partial clients to be available at each round. We compare FedCOG with FedAvg (McMahan et al., 2017), FedProx (Li et al., 2020b), MOON (Li et al., 2021a) here and report the results in Table 10. From the table, we see that 1) as the participation rate increases, the performance of all methods increase since more clients are available at each round. 2) FedCOG consistently outperforms the other methods across all participation rates. Specifically, FedCOG outperforms the second-best method (FedProx (Li et al., 2020b)) by $14\%$ relatively under $0.2$ participation rate.

Table 10: Effects of partial client participation rate. We consider $K = 50$ clients in total and change the sample rate for each round. FedCOG consistently performs the best for different participation rates.

| $K = 50$ | 0.1 | 0.2 | 0.3 | 0.4 |
|---|---|---|---|---|
| FedAvg | 18.64 | 37.44 | 41.47 | 46.73 |
| FedProx | 20.42 | 38.36 | 42.38 | 47.42 |
| MOON | 20.84 | 35.55 | 41.03 | 46.57 |
| **FedCOG** | **21.36** | **43.44** | **48.28** | **48.37** |

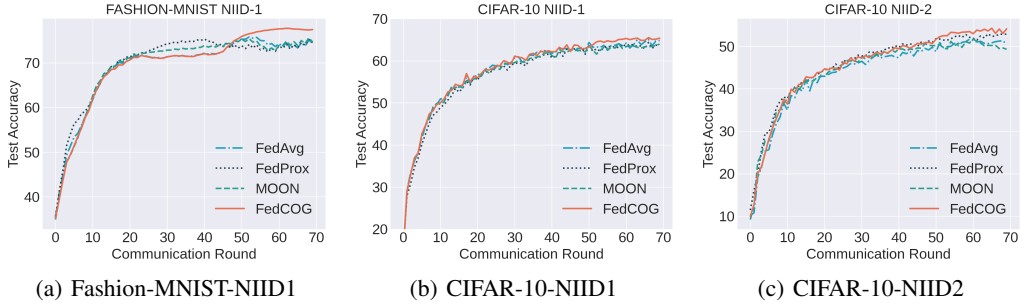

(a) Fashion-MNIST-NIID1  (b) CIFAR-10-NIID1  (c) CIFAR-10-NIID2

Figure 6: Visualization of convergence curves. We see that performance saturation after 70 rounds for most methods. Despite only launching 20 rounds of FedCOG (FedAvg for initial 50 rounds), our method still performs the best.

**Client number.** We investigate the effect of the number of clients for participation for CIFAR-10, see results in Table 11. Our method consistently outperforms all other approaches across different numbers of clients.

Table 11: Effects of client number $K$. FedCOG consistently performs the best for different client numbers.

| $K$ | 5 | 10 | 20 | 30 |
|---|---|---|---|---|
| FedAvg | 61.00 | 61.77 | 57.51 | 58.03 |
| FedProx | 61.28 | 61.90 | 57.08 | 58.98 |
| MOON | 60.64 | 60.41 | 56.25 | 58.35 |
| **FedCOG** | **61.72** | **61.98** | **57.67** | **59.14** |

## A.7 VISUALIZATION OF CONVERGENCE CURVES

Here, we visualize curves of different methods, where x-axis denotes the round index and the y-axis denotes the test accuracy of global model. In this experiment, our method is based on intial 50 rounds of FedAvg and 20 rounds of FedCOG to see a clearer improvement brought by FedCOG. From Figure 6, we see that 1) after 70 rounds of communication, the performance of global models of all methods saturate after 70 rounds of communication. 2) We see that our method consistently performs the best.

Please note that at this case, we are more interested at the performance improvement brought by FedCOG over FedAvg, which is significant through the figures. FedCOG can be further integrated with other methods such as FedProx Li et al. (2020b) and MOON Li et al. (2021a); please refer to Table 3.

