# OpenReview forum: "Fake It Till Make It: Federated Learning with Consensus-Oriented Generation"
_ICLR.cc/2024/Conference — ICLR 2024 poster_

### Official Review · Reviewer_3E8Z · 2023-10-31

**Soundness:** 3 good
**Presentation:** 3 good
**Contribution:** 2 fair
**Rating:** 6
**Confidence:** 4

**Summary:**

This paper proposed FedCOG, a synthetic data-assisted federated learning system, to mitigate the data heterogeneity in the training. The design mainly focused on the local training part. In the local training part, FedCOG first generates task-specific and client-specific data, and then uses knowledge distillation to train the local model. The experiment on computer vision benchmark datasets demonstrates that FedCOG performs well compared to existing FL baselines.

**Strengths:**

1. The proposed method is easy to follow. Different from other synthetic-data based methods, the FedCOG proposed task-specific and client-specific data for generation, which is novel and practical.

2. The paper is well-structured. The experiments include several existing FL baselines. The usage of the real-world FL multilabel dataset FLAIR is very rare in the FL literature.

**Weaknesses:**

1. I am confused about the data generation part of the reading. To my understanding, FedCOG took a learnable parameter for the data generator. What is the structure of the data generator? Does FedCOG update the weight of the data generator during the training as well? Could the author address more about how the data is generated locally?

2. In the experiment part, what are the sample numbers of the synthetic data in your setup?

3. The client number is so limited for the experiment related to standard datasets.

4. I am concerned that none of the selected baselines is a synthetic data-based method. I see the paper cites FedGen in the related work section. Why does the author not compare with the recent synthetic data-based methods such as FedGen[1] and DynaFed[2]?

[1]. Zhu, Zhuangdi et al. “Data-Free Knowledge Distillation for Heterogeneous Federated Learning.” Proceedings of machine learning research 139 (2021): 12878-12889 .

[2]. Pi, Renjie et al. “DYNAFED: Tackling Client Data Heterogeneity with Global Dynamics.” 2023 IEEE/CVF Conference on Computer Vision and Pattern Recognition (CVPR) (2022): 12177-12186.

**Questions:**

1. How would the FedCOG be harmonized with FedProx? I am curious about how the FedCOG does the local proximal term in the KD-based model training?

2. In Table 4, why FedProx took longer local training time compared to the FedCOG?

---

> ### Author Response · Authors · 2023-11-16
>
> Thank you for your time and valuable suggestions. Here are our detailed responses.
>
> ---
>
> **W1-1:** What is the structure of the data generator? Could the author address more about how the data is generated locally?
>
> **Response:** Sorry for the confusion.
>
> Actually, for computational efficiency, we do not introduce a data generator. Rather, the only learnable parameters are the generating inputs themselves. To be more specific, suppose we want to generate N samples and H,W,C are the height, width, and number of channels. The learnable parameter is a tensor with size of N\*H\*W\*C. To generate data, **only the inputs are set to be learnable** while the models are kept fixed.
>
> We also have explored the effects of introducing a generator (a linear layer with hidden size 8192, a upsampling and three 3x3 convolution layers) in Appendix (bottom of page 17), where we find that the achieved performance is comparable. Thus, we do not introduce a generator as it is more computation-efficient.
>
> **W1-2:** Does FedCOG update the weight of the data generator during the training as well?
>
> **Response:** No, the parameters related to data generation are fixed during training the local model; also see on the right part of Figure 1.
>
> ---
>
> **W2:** What are the sample numbers of the synthetic data in your setup?
>
> **Response:** The sample number of the synthetic data in our setup is 256. Please also note that this number is significantly smaller than the sample number of real data. For example, when the client number is 10 for CIFAR-10, each client has 5,000 samples on average.
>
> ---
>
> **W3:** The client number is limited.
>
> **Response:** Please note that we have shown these experiments in Table 7 and Table 8 in the Appendix with different client numbers, including 5, 10, 20, 30, 50.
>
> ---
>
> **W4:** None of the selected baselines is a synthetic data-based method? Why does the author not compare with methods such as FedGen?
>
> **Response:** Thanks for the advice. Actually, **FedReg (ICLR 2022) [3] in Table 1 is a synthetic data-based method**, where we can see that our method achieves significantly better performance.
>
> Following your advice, we now include comparison between with FedGen (ICML 2021) in Table 1 in the revision. Here, we show the comparison between FedCOG and FedGen below for convenience.
>
> From the table, we see that **FedCOG outperforms FedGen** across different datasets and heterogeneity types. This could result from the fact that FedCOG modifies the dataset itself to adjust the training of the whole model, which more fundamentally addresses the issue of data heterogeneity, while FedGen generates features to adjust the last layer of model only. Beside performance improvement, another key advantage of FedCOG is that FedCOG is compatible with Secure Aggregation while FedGen is not, because in FedGen the server needs to access each individual local model.
>
> [**Table R1.** Comparison between FedCOG and FedGen.]
> |   Method   | FMNIST-1  | FMNIST-2  | CIFAR10-1 | CIFAR10-2 | CIFAR100-1 | CIFAR100-2 |    Avg    |
> |:--:|:--:|:--:|:--:|:--:|:---:|:-:|:--:|
> |   FedGen   |   72.64   |   61.34   |   62.45   |   49.12   |   37.99    |   26.93    |   51.75   |
> | **FedCOG** | **77.34** | **73.68** | **64.83** | **54.00** | **42.88**  | **34.80**  | **57.92** |
>
> [3] Xu et al., Acceleration of Federated Learning with Alleviated Forgetting in Local Training, ICLR 2022.
>
> ---
>
> **Q1:** 1. How would the FedCOG be harmonized with FedProx? Curious about how FedCOG does the local proximal term in the KD-based training?
>
> **Response:** FedCOG can be seamlessly harmonized with FedProx. Take a conventional classification task as an example.
>
> FedCOG has two loss terms, which are both applied at the logit level (one for cross-entropy loss and one for KD-based loss), while FedProx introduces a model-level loss, which calculates element-wise l2 distance between local and global models. Thus, these **three loss terms** can be applied together to compute the loss and backward to update local model.
>
> ---
>
> **Q2:** In Table 4, why FedProx took longer local time compared to FedCOG?
>
> **Response:**
> The reason is that during local training, FedProx needs to launch two models, while FedCOG only needs to launch one model.
>
> In FedCOG, the soft labels of generated data are obtained by a one-time inference of the global model, which is fast. After this, during training, **only one model** (the local model) needs to be launched.
>
> In FedProx, **one local model and one global model** need to be launched during local training because proximal term in FedProx is computed by computing element-wise distance between local and global model. This cost will be larger, especially when the model size is larger.
>
> Please note that for FedCOG, the generation time, time for inferring soft labels and training time are all included in Table 4, making it a fair comparison.
>
> ---
>
> Overall, we hope that our responses can fully address your concerns and will be grateful for any feedback.

---

> > ### Comment · Reviewer_3E8Z · 2023-11-19
> >
> > Thank you for the effort in the response. I have some following questions regarding about it.
> >
> > Q2: I have a question regarding the "the soft labels of generated data are obtained by a one-time inference of the global model". At the 4th line in Algorithm 1, FedCOG performs complementary data generation in every round of the training. Do you mean the data generation in FedCOG is faster than the proximal term in FedProx?
> >
> > Additional Q1: Does the FedCOG only limit in training image modality? Would the FedCOG work on other data modality such as audio or language?

---

> > > ### Author Response · Authors · 2023-11-20
> > > **Thanks for your feedback!**
> > >
> > > We sincerely appreciate your feedback and is glad to further address your questions.
> > >
> > > ---
> > >
> > > **Q2:** I have a question regarding the "the soft labels of generated data are obtained by a one-time inference of the global model". Do you mean the data generation in FedCOG is faster than the proximal term in FedProx?
> > >
> > > **Response:** Sorry for the confusion. We mean at each round, the soft labels of generated data are obtained by a one-time inference of the global model, indicating that the global model is no longer needed during local model training. Thus, FedCOG only needs to launch one local model during training while FedProx needs to launch two models during training.
> > >
> > > Regarding the second question, the answer is yes. In our experiments, the generation process is fast (0.81 seconds), which makes our method faster than FedProx in a fair comparison.
> > >
> > > ---
> > >
> > > **Additional Q1:** Does the FedCOG only limit in training image modality? Would the FedCOG work on other data modality such as audio or language?
> > >
> > > **Answer:** That is a good point. Though we primarily focus on vision tasks to verify our idea of tackling data heterogeneity from data perspective, our method can also extend to other input modalities. Here, we elaborate this point on two types of modalities.
> > >
> > > (1) For modalities such as continuous signals and images where the raw inputs can be optimized by gradient-based optimization, the pipeline is exactly the same as illustrated in this paper, since we can optimize the inputs in an end-to-end manner.
> > >
> > > (2) For modalities such as text, where the raw inputs are harder to be directly optimized, it can be achieved with a slight difference. Since the raw inputs cannot be optimized via end-to-end gradient-based optimization, we do not optimize/generate raw inputs (which is discrete) but optimize/generate the intermediate embeddings (which is continuous). For example, suppose the overall pipeline of the model is: [a] discrete inputs are transformed into continuous embeddings after going through tokenizing and embedding layers, [b] continuous embeddings go through the downstream task model, [c] the model gives outputs. Then, our data generation process is generating the continuous embeddings at step [b], which are regarded as consensus data.
> > >
> > > Here, we implement our method on text-domain task on Yahoo! Answers [1] using a TextCNN model, where the other experimental settings remain the same. For data generation, we generate the intermediate features as the consensus data. Due to limited time and resource, we only apply our method on FedAvg and do not tune any hyper-parameter. And we only launch 5 rounds of FedCOG over FedAvg.
> > >
> > > From the table, we see that our method still outperforms the baseline, indicating the effectiveness of our metho on language data. Please also note that this result can be potentially further improved by launching more rounds of FedCOG and further hyper-parameter tuning.
> > >
> > > [**Table R2.** Results on text dataset.]
> > > | Setting | FedAvg | FedAvg+FedCOG |
> > > |:-------:| :----: | :-----------: |
> > > | Yahoo! Answers NIID-1  | 47.29  | **48.33**     |
> > >
> > > [1] Zhang, Xiang, Junbo Zhao, and Yann LeCun. "Character-level convolutional networks for text classification." Advances in neural information processing systems 28 (2015).
> > >
> > > ---
> > >
> > > We hope that our responses can fully address your concerns and will be grateful for any feedback.

---

> > > > ### Comment · Reviewer_3E8Z · 2023-11-20
> > > >
> > > > Thanks! I have raised up my score.

---

> > > > > ### Author Response · Authors · 2023-11-21
> > > > >
> > > > > We are so grateful for your recognition. Thanks for your time and your valuable advice!

---

### Official Review · Reviewer_SDr1 · 2023-11-02

**Soundness:** 4 excellent
**Presentation:** 3 good
**Contribution:** 3 good
**Rating:** 6
**Confidence:** 3

**Summary:**

This paper introduced a novel consensus scheme based on data generation to solve data heterogeneity problems in federated learning. It achieved a relatively higher accuracy on four public datasets with different degrees of heterogeneity. In the context of each individual client, the present study implemented a methodology wherein the global mode and local model which is extracted from the previous epoch were frozen. The objective was to train the generated data with the aim of optimizing the disparity between predictions made by the global model and those made by the local model, all the while mitigating any potential impact on the overall accuracy of the global model. All goals are evaluated on the generated dataset. Unlike current works focusing on the model, this paper provides a novel perspective from the local dataset. By enhancing the distribution of the local dataset, it claims to achieve better convergence. It achieved relatively higher accuracy on public datasets (FLAIR, Fashion-MNIST, CIFAR-10, and CIFAR-100) with different degrees of heterogeneity compared with federated learning (FL) algorithms like FedProx.

**Strengths:**

- The novel proposed method has a low overhead on each client, making it easy to apply in general FL tasks.
- Extensive experiments have been done to prove the advantages of the applied method.
- The paper is well organized and it's easy to follow.

**Weaknesses:**

- As far as I know, an enormous amount of work has proved that in the vision task, introducing unbalanced label distribution will influence the performance of the global model, and according to the results of the experiments, it's possible that this empirical idea is true. For details, please refer to the detailed comment C1.
- Evaluation is not strong enough; For details, please refer to the detailed comment C1~C3.
- No analysis of convergence is provided. For details, please refer to the detailed comment C3.

Detailed comments:
- C1 In the experiment results, the final accuracy on CIFAR-10 is relatively low, please try some more complicated networks other than the 5-layer CNN.
- C2 It's possible the network is not converged. To eliminate such a possibility, please provide a graph depicting the trend of convergence with the number of rounds on the server side.
- C3 What's more, the proposed method only achieved a little improvement in accuracy, it's not sure whether it's caused by insufficient experiments, please repeat and provide mean and standard error for all results.
- C4 We kindly request further experimentations involving the generation of datasets of varying sizes, with corresponding meticulous documentation of the associated overhead. Furthermore, if feasible, we encourage experimentation on datasets comprising high-resolution images uniformly, to enhance the comprehensiveness of the analysis.
- C5 Please add proofs for the convergence analysis. If possible, please add a formal security analysis to your method.

**Questions:**

1. This paper introduced data distribution from other clients, will this cause privacy leakage, making it easier for the attacker to learn data information from the clients?

2. Will generating new data for each client be identical to amplifying the global weight update direction collected in the last epoch?

---

> ### Author Response · Authors · 2023-11-16
>
> Thank you for your time and valuable suggestions. Here are our detailed responses.
>
> ---
>
> **W1:** Please try some more complicated networks other than the 5-layer CNN.
>
> **Response:** Thanks for the advice. Here, we run experiments on ResNet18. To demonstrate the effectiveness of our method, we only run one round of FedCOG.
>
> From the table below, we see that the accuracy on NIID-1 is much higher now while the accuracy on NIID-2 (more heterogeneous) is still similar to previous results. Nevertheless, our method still demonstrates evident effectiveness.
>
> [**Table R1.** Results on larger models.]
> | Setting | FedAvg | FedAvg+FedCOG | FedProx | FedProx+FedCOG |
> |:-------:| :----: | :-----------: | :-----: | :------------: |
> | NIID-1  | 77.64  | **78.34**     | 76.06   | **77.39**      |
> | NIID-2  | 44.81  | **48.34**     | 54.08   | **55.50**      |
>
> ---
>
> **W2:** Please provide a graph depicting the trend of convergence with the number of rounds on the server side.
>
> **Response:** Thanks for your advice, we include the convergence figures in Figure 6 in the revision. We show that the network has converged, and our method achieves the best performance.
>
> ---
>
> **W3:** Please repeat and provide mean and standard error.
>
> **Response:** Thanks for the advice. Please note that:
>
> (1) In Table 1, we have reported mean and standard error, which shows that our method can significantly and consistently outperform other methods.
>
> (2) Sorry for the confusion. For other results, the reported values are mean values of 3-5 trials, and we did not include standard error due to space limits, such that we could show more comparisons.
>
> Here we list several examples.
>
> [**Table R2.** Mean and std.]
> | Method |     FMNIST-1     |     FMNIST-2     |    CIFAR10-1     |    CIFAR10-2     |
> |:------:|:----------------:|:----------------:|:----------------:|:----------------:|
> | FedAvg | 73.07 $\pm$ 0.08 | 64.11 $\pm$ 0.78 | 64.36 $\pm$ 0.11 | 50.55 $\pm$ 0.45 |
> | FedCOG | **77.34** $\pm$ 0.07 | **73.68** $\pm$ 0.38 | **64.83** $\pm$ 0.12 | **54.00** $\pm$ 0.84 |
>
> ---
>
> **W4-1:**  Request further experimentations involving the generation of datasets of varying sizes, with corresponding meticulous documentation of the associated overhead.
>
> **Response:** Thanks for the helpful advice! We now include a table to demonstrate the relationship between generation time and number of generated samples in Table 11 in the revision. For convenience, we also put the table here. We can see that the generation time is little (we generate 256 samples throughout the paper), indicating the efficiency of our method. As a reference, the conventional local training takes 14.81 seconds.
>
> [**Table R3.** Generation time (seconds) v.s. number of generated samples.]
> | Number of generated samples  | 16   | 32   | 64   | 128  | 256  | 512  | 1024 |
> | ---- | ---- | ---- | ---- | ---- | ---- | ---- | ---- |
> | Generation time (s) | 0.72 | 0.78 | 0.79 | 0.80 | 0.81 | 0.83 | 1.27 |
>
> **W4-2:** If feasible, we encourage experimentation on datasets comprising high-resolution images.
>
> **Response:** Please note the images in the FLAIR dataset (Table 2) have a high resolution of 256\*256.

---

> ### Author Response · Authors · 2023-11-16
>
> **W5:** Please add proofs for the convergence analysis. If possible, please add a formal security analysis to your method.
>
> **Response:**
>
> Here, we provide convergence analysis and some interpretations.
>
> **1. Convergence:**
>
> Please refer to Section A.6 in the Appendix in the revision. We show that our method has a **theoretical convergence guarantee besides achieving pleasant performance**.
>
> **2. Interpretations:**
>
> From standard FL theoretical literature, it is well-known that lower dissimilarity contributes to faster convergence both empirically and theoretically [2,3]. At the same time, our method contributes to **lower model difference**, which corresponds to lower dissimilarity among clients in the standard FL theoretical literature (e.g., bounded dissimilarity in [1,2]); please refer to evidence in Figure 2 (a).
>
> These two key observations can provide implications and evidence for the convergence of our method.
>
> Please also note that the focus of our paper is to improve the performance of federated learning from a practical view and our contribution lies on proposing a new attempt on tackling data heterogeneity but not theory. Thus, we did not include a convergence analysis previously.
>
> [1] Karimireddy, Sai Praneeth, et al. "Scaffold: Stochastic controlled averaging for federated learning." International conference on machine learning. PMLR, 2020.
>
> [2] Wang, Jianyu, et al. "Tackling the objective inconsistency problem in heterogeneous federated optimization." Advances in neural information processing systems 33 (2020): 7611-7623.
>
> [3] Li, Tian, et al. "Federated optimization in heterogeneous networks." Proceedings of Machine learning and systems 2 (2020): 429-450.
>
> ---
>
> **Q1:** This paper introduced data distribution from other clients, will this cause privacy leakage, making it easier for the attacker to learn data information from the clients?
>
> **Response:** Sorry for the potential confusion. We would like to emphasize that **our method does not introduce data distribution from other clients, which does not cause privacy leakage**. The shared information is exactly the same as FedAvg: the local model. No information about distribution is required for sharing.
>
> ---
>
> **Q2:** Will generating new data for each client be identical to amplifying the global weight update direction collected in the last epoch?
>
> **Response:** We believe that generating new data for each client is not equivalent to amplifying the global weight update direction. The reasons are twofold. On one hand, this can be supported by the results of FedAvgM, which introduces momentum-based global update for each round. On the other hand, we keep the number of local model updates the same for all methods, making sure that all methods are with the same update scale.
>
> ---
>
> Overall, we hope that our responses can fully address your concerns and will be grateful for any feedback.

---

> ### Author Response · Authors · 2023-11-21
>
> Dear Reviewer:
>
> Thanks again for the valuable comments.
>
> We have now provided more clarifications, explanations, and experiments to address your concerns. We also included the convergence analysis as requested by you and followed the advice of all reviewers to improve our paper.
>
> Please kindly let us know if anything is unclear. We truly appreciate this opportunity to improve our work and shall be most grateful for any feedback you could give to us.

---

> ### Author Response · Authors · 2023-11-22
> **We sincerely anticipate your feedback as the Discussion stage will end in 18 hours.**
>
> Dear Reviewer,
>
> We have followed your advice to improve our work including more experiments and convergence analysis. As Discussion Stage will end in 18 hours, we would be grateful if you could check our responses and reconsider your rating.
>
> Best regards,
> Authors

---

> > ### Comment · Reviewer_SDr1 · 2023-11-23
> >
> > I appreciate the author's response and adjusted my rating accordingly.

---

### Official Review · Reviewer_wN99 · 2023-11-03

**Soundness:** 3 good
**Presentation:** 3 good
**Contribution:** 3 good
**Rating:** 6
**Confidence:** 4

**Summary:**

This paper proposed FedCOG, a Federated Learning (FL) algorithm that facilitates learning via augmented data generated from the global model, which is later used for knowledge distillation between the global and client models. This scheme is compatible with most existing FL algorithms. Its effects have been empirically verified on real-world datasets.

**Strengths:**

\+ This paper tackles a crucial challenge in FL which is Data heterogeneity. Their core idea of data correction for achieving global data consensus is well-motivated.

\+ Data generation by capturing the residual knowledge between the global and the client model is novel.

\+ Sensitivity analysis is well designed and conducted.

\+ This paper is clearly written. Related work is comprehensive.

**Weaknesses:**

\- Data generation on the client step brings extra computation workload compared with classic FL or FL with data generation on the server side.

\- The method seems to be designed purely for vision tasks. Authors could discuss if the proposed method can be extended to scenarios with other input modalities, such as text inputs.

\- This paper would further benefit from theoretical derivations to interpret why generated data on the client side helps in improving global model performance.

\- Concerns on Experiments: All methods in Table 1 achieve notably lower accuracies than SOTA FL methods. I am concerned that the model arch, communication round, or optimizer setting is not well set up for appropriate comparison.

**Questions:**

\- Since tackling data heterogeneity is the key of this paper, I suggest authors conduct more experiments on data with Dirichlet distribution by varying the hyper-parameter $\beta$. More results with changing heterogeneities would further validate the effects of the proposed methods.

---

> ### Author Response · Authors · 2023-11-16
>
> Thank you for your time and valuable suggestions. Here are our detailed responses.
>
> ---
>
> **W1:** Data generation on the client step brings extra computation workload compared with classic FL or FL with data generation on the server side.
>
> **Response:** Thanks for you comments. We would like to response from three aspects.
>
> 1. **Computation efficiency.** Our method introduces **minor** computation workload compared with FedAvg and is more efficient than many FL baselines. To be more specific, on CIFAR-10, the generation only takes **0.81 seconds**, while as a reference the conventional SGD-based training takes 14.81 seconds (FedAvg). Compared with other baselines, our method only needs to launch one model during local training while baselines such as FedProx needs to launch two models and MOON needs to launch three models.
>
> 2. **Compatibility with standard FL protocols.** Our method is compatible with **secure aggregation** [1]. In practice, the secure aggregation is required to further protect the communicated model parameters such that the server only obtains the aggregated information (i.e., model). However, those methods that generate data on the server side require that the server obtains each specific local model from client, making them incompatible with secure aggregation technique. In contrast, the server in our method only needs to obtain the aggregated model, making it compatible with secure aggregation. This is critical as the local model itself may reveal some sensitive information and thus should not be overlooked.
>
> 3. **Communication and privacy are two first-order concerns** in FL as pointed by the influential and pioneering paper [2]. Following this spirit, we decided to introduce minor computation cost for improving algorithm utility, which is unavoidable [3].
>
> [1] Bonawitz, Keith, et al. "Practical secure aggregation for privacy-preserving machine learning." proceedings of the 2017 ACM SIGSAC Conference on Computer and Communications Security. 2017.
>
> [2] Kairouz, Peter, et al. "Advances and open problems in federated learning." Foundations and Trends® in Machine Learning 14.1–2 (2021): 1-210.
>
> [3] Zhang, Xiaojin, et al. "Trading off privacy, utility and efficiency in federated learning." ACM Transactions on Intelligent Systems and Technology (2022).
>
> ---
>
> **W2:** Authors could discuss if the proposed method can be extended to scenarios with other input modalities, such as text inputs.
>
> **Response:** That is a good point. Though we primarily focus on vision tasks to verify our idea of tackling data heterogeneity from data perspective, our method can also extend to other input modalities. Here, we elaborate this point on two types of modalities.
>
> (1) For modalities such as continuous signals and images where the raw inputs can be optimized by gradient-based optimization, the pipeline is exactly the same as illustrated in this paper, since we can optimize the inputs in an end-to-end manner.
>
> (2) For modalities such as text, where the raw inputs are harder to be directly optimized, it can be achieved with a slight difference. Since the raw inputs can not be optimized via end-to-end gradient-based optimization, we do not optimize/generate raw inputs (which is discrete) but optimize/generate the intermediate embeddings (which is continuous). For example, suppose the overall pipeline of the model is: [a] discrete inputs are transformed into continous embeddings after going through tokenizing and embedding layers, [b] continuous embeddings go through the downstream task model, [c] the model gives outputs. Then, our data generation process is generating the continuous embeddings at step [b], which are regarded as consensus data.

---

> ### Author Response · Authors · 2023-11-16
>
> **W3:** This paper would further benefit from theoretical derivations to interpret why generated data on the client side helps in improving global model performance.
>
> **Response:**
>
> Here, we provide some interpretations and a theoretical convergence analysis of our method.
>
> **1. Interpretations:**
>
> From standard FL theoretical literature, it is well-known that lower dissimilarity (between clients) contributes to faster convergence both empirically and theoretically [2,3]. At the same time, our method contributes to lower model difference, which corresponds to lower dissimilarity among clients in the standard FL theoretical literature (e.g., bounded dissimilarity in [1,2]); please refer to evidence in Figure 2 (a). These two key observations can provide implications and evidence for the convergence of our method.
>
> **2. Convergence:**
>
> Please refer to Section A.6 in the Appendix in the revision. We show that our method has a **theoretical convergence guarantee** besides achieving pleasant performance.
>
> Please also note that the focus of our paper is to improve the performance of federated learning from a practical view and our contribution lies on proposing a new attempt on tackling data heterogeneity but not theory. Thus, we did not include a convergence analysis previously.
>
> [1] Karimireddy, Sai Praneeth, et al. "Scaffold: Stochastic controlled averaging for federated learning." International conference on machine learning. PMLR, 2020.
>
> [2] Wang, Jianyu, et al. "Tackling the objective inconsistency problem in heterogeneous federated optimization." Advances in neural information processing systems 33 (2020): 7611-7623.
>
> [3] Li, Tian, et al. "Federated optimization in heterogeneous networks." Proceedings of Machine learning and systems 2 (2020): 429-450.
>
> ---
>
> **W4:** I am concerned that the model arch, communication round, or optimizer setting is not well set up for appropriate comparison.
>
> **Response:** We would like to emphasize that we follow exactly the same setups as [1,2] including model architecture (CNN), optimizer (SGD), and learning rate (0.01). We set the number of rounds as 70 because we find that after 70 rounds the performance remains nearly the same as the 70th round's.
>
> The reason why the performance seems low on average is that many methods are not robust towards diverse settings (e.g., datasets and heterogeneity types). For example, SCAFFOLD achieves 4% higher than FedAvg on NIID-1 of Fashion-MNIST but 6% lower than FedAvg on NIID-2 of Fashion-MNIST.
>
> [1] Li, Qinbin, Bingsheng He, and Dawn Song. "Model-contrastive federated learning." Proceedings of the IEEE/CVF conference on computer vision and pattern recognition. 2021.
>
> [2] Luo, Mi, et al. "No fear of heterogeneity: Classifier calibration for federated learning with non-iid data." Advances in Neural Information Processing Systems 34 (2021): 5972-5984.
>
> ---
>
> **Q1:** I suggest authors conduct more experiments on data with Dirichlet distribution by varying the hyper-parameter beta.
>
> **Response:** Thanks for the advice. Actually, we have included such experiments in Figure 3 (a). Here, we show the results again for convenience. We see that our proposed FedCOG consistently performs the best at different beta.
>
> [**Table R1.** Effects of hyper-parameter beta.]
> | Beta       | 0.1       | 0.5       | 1.0       | 5.0       |
> | ---------- | --------- | --------- | --------- | --------- |
> | FedAvg     | 63.50     | 64.51     | 66.55     | 67.70     |
> | FedProx    | 63.00     | 63.94     | 67.29     | 67.01     |
> | MOON       | 63.28     | 64.09     | 64.07     | 67.78     |
> | **FedCOG** | **64.46** | **65.87** | **67.99** | **69.40** |
>
> ---
>
> Overall, we hope that our responses can fully address your concerns and will be grateful for any feedback.

---

> ### Author Response · Authors · 2023-11-21
>
> Dear Reviewer:
>
> Thanks again for the valuable comments.
>
> We have now provided more clarifications, explanations, and experiments to address your concerns and followed the advice of all reviewers to improve our paper.
>
> Please kindly let us know if anything is unclear. We truly appreciate this opportunity to improve our work and shall be most grateful for any feedback you could give to us.

---

### Official Review · Reviewer_iXGo · 2023-11-09

**Soundness:** 3 good
**Presentation:** 3 good
**Contribution:** 3 good
**Rating:** 6
**Confidence:** 3

**Summary:**

This paper focuses on federated learning (FL) in the presence of data heterogeneity. Different from the existing methods which usually consider this data heterogeneity as an inherent property and attempt to mitigate the adverse effects, this paper proposes to handle the heterogeneity by generating new data, called FedCOG. There are two key components in FedCOG, including complementary data generation and knowledge-distillation-based model training. It can be plug-and-play, and naturally compatible with standard federated learning protocols.  Extensive experiments on classical and real-world datasets proved the effectiveness of the proposed method.

**Strengths:**

1. The proposed method is novel, handling the data heterogeneity from the perspective of the data generation, instead of correcting the model.
2. The proposed method can be a plug-and-play model, and it is naturally compatible with standard FL protocols.

**Weaknesses:**

1. The motivation for why we need to use data generation, instead of recent popular methods based on model correction, is somewhat not clear. When the training dataset is very large, the proposed method therefore needs to generate a large amount of data in order to achieve alignment, which is costly, then it seems like model correction is a better choice in such a scenario.
2. As the paper mentioned, the proposed method FedCOG has two advantages, i.e., plug-and-play and compatibility with standard FL protocols, however, the unique advantages of this data generation method, compared to previous model correction methods, are still ambiguous. Could you please elaborate more regarding them?

**Questions:**

(see above)

---

> ### Author Response · Authors · 2023-11-16
>
> Thank you for your time and valuable suggestions. Here are our detailed responses.
>
> ---
>
> **W1-1:** The motivation for why we need to use data generation, instead of recent popular methods based on model correction, is somewhat not clear.
>
> **Response:** Sorry for the confusion. Our motivations are twofold.
>
> 1. **Heterogeneous data is the fundamental cause of performance degradation while model-correction-based methods do not directly operate on data.** Previous model correction methods focus on remedying the negative effects of data heterogeneity by model correction techniques while leaving the dataset as heterogeneous as before. While data is the root of performance degradation, this motivates us to deal with the heterogenous datasets themselves to more fundamentally address the issue of data heterogeneity. We believe that this will be an interesting future direction, especially because there are more and more advanced data generation techniques.
> 2. **Merely relying on correction-based methods cannot fully address data heterogeneity,** which motivates us to explore a new direction that addresses this from an orthoganal perspective. We expect to see that in the future the issue of data heterogeneity can potentially be organically addressed by integration of techniques from diverse perspectives.
>
> **W1-2:** The proposed method needs to generate more data when training dataset is large?
>
> **Response:** Sorry for the potential misunderstanding we have caused. Actually, the generated number does not scale up with the number of training samples. Throughout the paper, **the generated number keeps the same (256)** no matter how many samples each client has (e.g., 10k, 5k). Note that the required generation time is only **0.81 seconds** (on CIFAR-10) since the only learnable parameters are the inputs while the models are fixed, and **only 100 steps** of optimization is sufficient.
>
> Here is the rationale. In FL, the number of local updates is strongly related to convergence, thus it is set to a fixed moderate number no matter how many data samples the client has. This means that for each round, the number of samples used for updating the model is the same no matter how many data samples the client has (number of local updates multiplied by batch size). Thus, the generated number for each round can also be set as the same number no matter how many data samples the client has.
>
> Besides, our method is **actually efficient** because our method only needs to launch **one model** during local model training (the soft labels from global model is obtained for only **one-time inference** and only the local model needs to be launched). On the contrary, for example, during local model training, FedProx needs to launch **two models** (one local model and one global model); MOON needs to launch **three models** (two local models and one global model); SCAFFOLD needs to launch **three models** (one local model, one local control variate and one global control variate).
>
> ---
>
> **W2:** The unique advantages of this data generation method, compared to previous model correction methods.
>
> **Response:** The unique advantage of our method is that our method not only more fundamentally alleviates the heterogeneity level of clients' data, but also enhance alignment among local models at the same time.
>
> FedCOG consists of two key parts: data generation and knowledge distillation, which organically alleviate the issue of data heterogeneity. (1) First, the generated consensus data can effectively decrease the difference among clients' datasets, therefore reducing the data heterogeneity level to more fundamentally alleviate this issue. (2) Second, based on the consensus data, we force each local model's outputs to align with global model's outputs, which functions to implicitly align local models like correction-based methods. Besides, unlike correction-based methods that need to launch several models, our method only needs to launch one local model during training as the soft labels of consensus data can be obtained through one-time inference, which is efficient.
>
> Please note that besides the technical contribution, our contribution also lies on exploring a **novel perspective of dealing with data heterogeneity**, which can potentially inspire more future works to effectively address data heterogeneity.
>
> ---
>
> Overall, we hope that our responses can fully address your concerns and will be grateful for any feedback.

---

> ### Author Response · Authors · 2023-11-21
>
> Dear Reviewer:
>
> Thanks again for the valuable comments.
>
> We have now provided more clarifications and explanations to address your concerns and followed the advice of all reviewers to improve our paper.
>
> Please kindly let us know if anything is unclear. We truly appreciate this opportunity to improve our work and shall be most grateful for any feedback you could give to us.

---

> > ### Comment · Reviewer_iXGo · 2023-11-21
> > **Response to authors**
> >
> > Thank you for the responses, which addressed my concerns, therefore, I updated my rating to 6.

---

> > > ### Author Response · Authors · 2023-11-22
> > >
> > > We are so grateful for your recognition. Thanks for your time and your valuable advice!

---

### Author Response · Authors · 2023-11-16
**Thanks for all reviewers' time and valuable comments.**

Dear Reviewers:

We would like to express our profound gratitude for your time and insightful comments.

We are truly delighted to note that all the reviewers acknowledge the **novelty of our method and the writing of our paper**. We sincerely apologize for any confusion our initial draft may have caused, and hopefully we have addressed them in our responses.

We have now included a detailed theoretical convergence analysis of our method, as suggested by Reviewers SDr1 and wN99. We kindly request you to refer to our theoretical results and proofs in the Appendix. Here, we demonstrate that our method not only offers performance improvements, but also provides a theoretical convergence guarantee.

Thank you once again for your valuable inputs.

Best Regards,

Authors

---

### Meta-Review · Area_Chair_qd9W · 2023-12-08

**Metareview:**

The paper presents an innovative approach to handling data heterogeneity from the perspective of data generation in the setting of federated learning.

During the rebuttal, some common strengths highlighted by reviewers include: 1) novel methods; 2) ease of application in general FL tasks; 3) extensive experiments; 4) clear writing.

The authors made significant clarifications during the rebuttal, and three reviewers acknowledged that all major concerns had been well addressed and subsequently raised their ratings.

Given the positive feedback from reviewers, the AC recommends accepting this paper.

To enhance the overall quality of the paper, the authors are strongly encouraged to revise their paper according to the reviewers' detailed feedback regarding any unclear claims in the data generation part, experiment setup, and convergence analysis.

**Justification For Why Not Higher Score:**

NA

**Justification For Why Not Lower Score:**

Although the paper lacks in-depth theoretical analysis, its strengths in innovation, wide applicability, and empirical validation, coupled with positive feedback from peers, affirm its quality and relevance, warranting the current score.

---

### Decision · Program_Chairs · 2024-01-16

Accept (poster)